# Assessment of plum rain's impact on power system emissions in Yangtze-Huaihe River basin of China

Guangsheng Pan[1,3], Qinran Hu [1,3], Wei Gu [1✉], Shixing Ding[2], Haifeng Qiu [1] & Yuping Lu[1]

As a typical climate that occurs in the Yangtze-Huaihe River basin of China with a size of 500,000 km$^2$, plum rain can reduce the photovoltaic (PV) potential by lowering the surface irradiance (SI) in the affected region. Based on hourly meteorological data from 1980 to 2020, we find that plum rain can lower the SI in the affected region with a weekly peak drop of more than 20% at the most affected locations. This SI drop, coupled with a large number of deployed PV systems, can cause incremental $CO_2$ emissions (ICEs) of local power systems by increasing the additional thermal power. Using a cost optimization model, we demonstrate that the ICEs in 2020 already reached 1.22 megatons and could range from 2.21 to 4.73 megatons, 3.47 to 7.19 megatons, and 2.97 to 7.43 megatons in 2030, 2040, and 2050, respectively, considering a change trend interval of a ±25% fluctuation in power generation and demand in the different years. To offset these ICEs, we compare four pathways integrated with promising technologies. This analysis reveals that the advanced deployment of complementary technologies can improve the PV utilization level to address climate impacts.

[1] School of Electrical Engineering, Southeast University, Nanjing, China 210096. [2] School of Cyber Science and Engineering, Southeast University, Nanjing, China 210096. [3]These authors contributed equally: Guangsheng Pan, Qinran Hu. ✉email: wgu@seu.edu.cn

To achieve the Paris Agreement goal of constraining the increase in the global average surface temperature[1,2], many countries worldwide are adopting effective measures to reduce greenhouse gas emissions. As the world's largest carbon producer, China has recently proposed a new carbon emission reduction target in the 14th Five-Year Plan[3]. According to this target, China strives to reach a carbon dioxide emission peak by 2030 and to achieve carbon neutrality by 2060. To realize this vision, the Chinese government has defined aggressive new energy development goals with a clear timetable[4]. Due to its advantages of easy deployment, either centralized or distributed as modular technology[5], and its cost competitiveness with grid electricity in China[6], photovoltaic (PV) systems will continue to steadily and rapidly grow in the future[7].

As an important product of the northward advancement of the East Asian summer monsoon, plum rain mainly occurs in the Yangtze–Huaihe River basin of China, affecting an area of nearly 500,000 km$^2$ [8]. During the plum rain period, continuous cloudy and rainy weather conditions prominently occur, which can easily cause floods, reduce crop yields, and affect people's transportation patterns. Additionally, clouds and precipitation during the plum rain period can reduce the surface irradiance (SI), yielding economic and carbon challenges to the operation of power systems by reducing the PV potential. The East Asian summer monsoon and El Niño-Southern Oscillation are important factors influencing the temporal and spatial changes in plum rain[9]. As a result, the duration and impact degree of plum rain in various years are quite different. Generally, plum rain lasts for approximately 20–30 days with accumulated precipitation ranging from 200–400 mm. However, the ultralong plum rain period in 2020 lasted longer than 40 days in most affected areas and even 2 months in local areas with accumulated precipitation exceeding 800 mm[10]. This ultralong plum rain period not only affected the lives of people but also attracted the attention of energy-related practitioners.

The impact of typical climate and environmental issues on the energy demand[11,12], renewable power generation[13–18], and energy systems[19,20] have been extensively studied. For instance, anthropogenic aerosol emissions and changes in cloud cover[13], and frequent extreme conditions[14] can reduce the PV potential. The warming of the Indian Ocean can lead to a secular decrease in the wind power potential in India[15]. Climate change can decrease the dry season hydropower potential, thus worsening the mismatch between the seasonal electricity supply and peak demand[17]. As a result, future energy systems dominated by renewables can face challenges in the reliability of energy supply. Therefore, it is necessary to deeply understand the negative impacts of typical climate conditions caused by potential factors before the large-scale implementation of renewable energy techniques. As a typical climate, the effect of plum rain on the energy system in the Jianghuai River Basin cannot be ignored. The provinces involved in the Jianghuai River Basin include 6 provinces and 1 municipality, which account for 28 and 35%, respectively, of the national electricity demand and gross domestic product. PV systems, as a booming power generation technology, have been installed with a capacity of 65.73 GW in these 6 provinces and 1 municipality by the end of 2020, accounting for 26% of the total installed PV capacity in China, at 253.43 GW[21]. However, the specific impact degree of plum rain on the abovementioned areas remains unclear. To our knowledge, this work is the first attempt to analyze the above impact degree.

This paper aims to assess the plum rain impact on $CO_2$ emissions of power systems in the Yangtze–Huaihe River basin of China. First, weekly year-average data of 41-year hourly radiation-surface incoming shortwave flux (RSISF) data from 1980 to 2020 with a spatial resolution of 1/2° latitude by 2/3°

longitude are used to observe the plum rain effects on the all-sky SI in the affected region (R1), which are compared to those in the unaffected surrounding region (R2). Then, a province-level analysis of the impact degree on SI is performed in each province or municipality. Second, based on a province-level clustered unit commitment (CUC) model, the plum rain impact on the incremental $CO_2$ emissions (ICEs) of province-level power grids with the incremental PV installed capacity from 2020 to 2050 is studied. Finally, several pathways are introduced to compare and analyze their economic and technical characteristics in regard to the reduction in the negative plum rain impact on power systems. The results indicate that plum rain can obviously reduce PV generation through SI weakening in R1 and thus can increase the $CO_2$ emissions of power systems. Before the deployment of more PV systems in R1 in the future, we should select regionally differentiated supporting facilities to offset the negative impact of plum rain on power systems.

## Results

**Plum rain obviously reduces the SI in R1**. The annual mean weekly SI curves over the period 1980–2020 in R1 and R2 (see Supplementary Fig. 1 for details) are displayed in Fig. 1a. The shaded part indicates the duration of the plum rain period, from June 18 to July 22 of a given year. As indicated, the SI in R1 exhibits a notable downward trend over that in R2 during the plum rain period. The largest decline occurred in the 26th week, with a drop of 27.6 W/m$^2$. Outside the plum rain period, the SI curves for these two regions overlapped very highly. Based on the above observations, it is clear that plum rain can obviously reduce the SI in R1. The impact degree $\theta P_{MEAN}$ of plum rain on the SI in each province or municipality from the 25th week to the 29th week, defined as the ratio of the difference between the mean SI in R2 and SI at a given location in R1 to the mean SI in R2 (see "Methods" for details), is shown in Fig. 1b. Overall, as time goes by, the impact degree of plum rain first increases and then decreases, with a peak at 26th week in Shanghai, Zhejiang, Jiangxi, Hunan, and Anhui. In the 26th week, Shanghai attains the highest impact degree of nearly 20%, followed by Zhejiang (17%) and Jiangxi (13%). Compared to the obvious peaks in the above provinces, the volatility in the other provinces is minor. Anhui and Nanjing exhibit an impact higher than 5% in 4 weeks, and the peak value is approximately 10%. In contrast, in Hunan and Hubei, the impact degree reaches about 5% in only 3 weeks, and the peak value remains below 8%. The regional distribution of the 26-week impact degree with a spatial resolution of 1/2° latitude by 2/3° longitude is further presented in Fig. 2. The intersection of Shanghai, Jiangsu, and Zhejiang exhibits the highest impact degree, reaching 21%. With this intersection area as the center, the impact degree gradually decreases towards the surrounding areas. Jiangsu, Anhui, and Hubei achieve the lowest impact degree among the areas bordering R2, even close to zero.

**Plum rain can cause megaton-scale ICEs**. The ICEs of provincial power systems due to plum rain in the different years are shown in Fig. 3 (see Supplementary Fig. 2 for the detailed impact process). Three cases with different power generation and demand change trends in the different years are defined. The base case refers to the national average change trends forecasted by the State Grid Energy Research Institute (Supplementary Table 1)[22]. On this basis, we consider a fluctuation interval of ±25% to cover the inconsistent change trends at the national and provincial levels. As indicated, the ICEs in 2020 already reached 1.22 megatons and could reach [2.21, 4.73], [3.47, 7.19] and [2.97, 7.43] in 2030, 2040, and 2050, respectively. The impact of the considered change trends on the ICEs increases year by year,

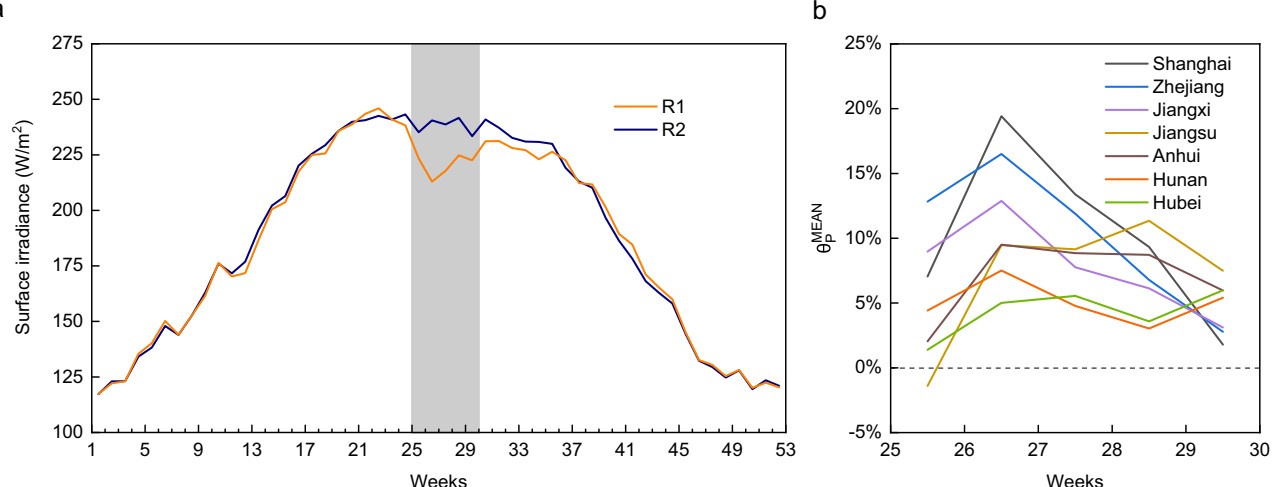

**Fig. 1 Plum rain effects on the surface irradiance (SI) in the affected region (R1). a** The 52-week SI within a year, where R2 denotes the unaffected surrounding region. **b** Impact degree of plum rain on the SI from the 25th week to the 29th week, where $\theta_{MEAN}^P$ denotes the mean impact degree of plum rain on the SI in each province or municipality.

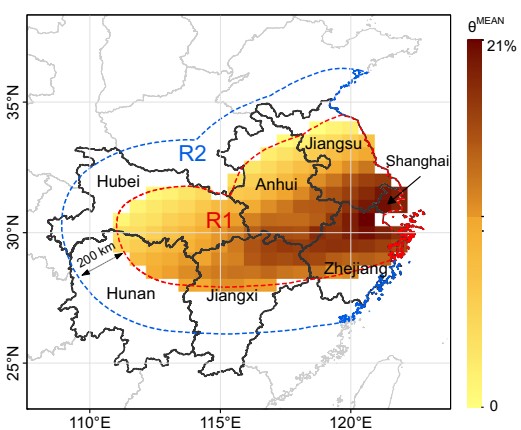

**Fig. 2 Spatial distributions of the impact degree within the affected region (R1).** The region enclosed within the red dashed line is R1. The region between the red and blue dashed lines is the unaffected surrounding region (R2). $\theta^{MEAN}$ denotes the impact degree of plum rain on the surface irradiance at a given location.

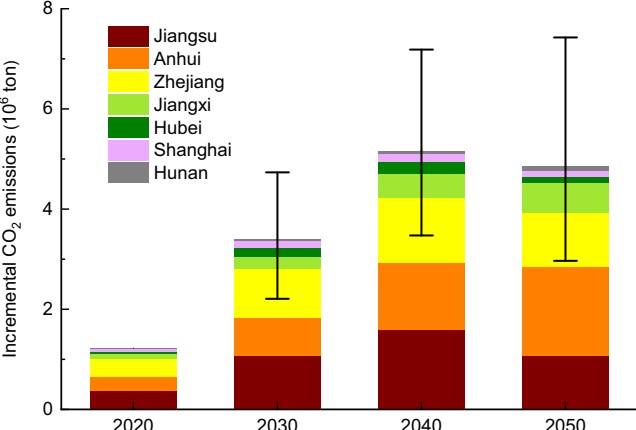

**Fig. 3 Plum rain effects on the incremental CO$_2$ emissions (ICEs) from 2020 to 2050.** The error bar defines the range of the ICEs based on the generation and demand change trends fluctuating by ±25% from 2030 to 2050 on the basis of the national average values.

reaching more than 50% in 2050. Specific to each province or municipality, the ICEs in the base case are presented. The ICEs in Jiangsu in 2020, 2030, and 2040 are the highest among all provinces, and they increase year by year with a peak value of 1.59 megatons in 2040, after which they notably drop in 2050. The second-highest emissions occur in Anhui, and its ICEs increase year by year and reach a maximum value of 1.76 megatons in 2050. Zhejiang, Hubei, and Shanghai rank third, fifth, and sixth, respectively, regarding their ICEs, and the variation tendency is the same as that in Jiangsu, whereas Jiangxi and Hunan rank fourth and seventh, respectively, regarding their ICEs, and the variation tendency is the same as that in Anhui. The comprehensive performance of the above provinces or municipalities results in the ICEs in the affected region first increasing and then decreasing in the base case, reaching a peak value in 2040.

We can explain the impact of plum rain on the ICEs in the different provinces or municipalities via the following two steps. The first step indicates that plum rain reduces PV potential in R1, which can be further divided into non-human and human factors. Non-human factors include impact degree and affected area ratio of plum rain, as well as PV capacity factor (CF) in each province

or municipality (Supplementary Table 2). Figure 4a shows the province-level daily generation reduction for a 1-kW solar panel, which is determined by non-human factors. As indicated, plum rain exerts the highest impact on PV generation in Shanghai, with a daily generation reduction of 0.43 kWh, followed by Zhejiang, Jiangsu, and Anhui, with a daily generation reduction above 0.2 kWh. As a comparison, Hunan, with the lowest affect area ratio of 18%, exhibits the lowest daily generation reduction with a value of 0.03 kWh. The human factor involves PV capacity allocation in the different provinces or municipalities. As of the end of December 2020, the total installed PV power capacity in Jiangsu, Anhui, Zhejiang, Jiangxi, Hubei, Shanghai, and Hunan reached 16.84, 13.70, 15.17, 7.76, 6.98, 1.37, and 3.91 GW, respectively. The above unit PV generation reduction and PV capacity determine the actual PV generation reduction caused by plum rain.

The second step involves the compensation of missing PV power generation, which is most likely satisfied through the easy scheduling of coal generators (CGs) and natural gas generators (NGs) in China (Fig. 4b). Owing to the relatively low fuel prices, CGs are mainly adopted to compensate for the PV generation

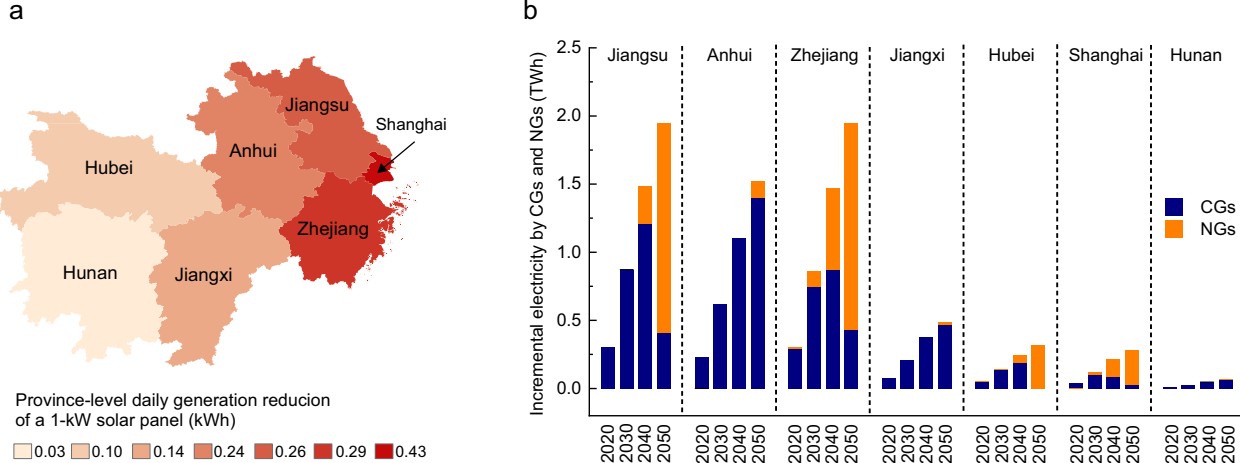

**Fig. 4 Analysis of the key factors influencing incremental CO$_2$ emissions caused by plum rain. a** Province-level daily generation reduction of a 1-kW solar panel. The darker the color, the higher the daily generation reduction is. **b** Plum rain effects on incremental electricity by coal generators (CGs) and natural gas generators (NGs) of different provinces or municipalities from 2020 to 2050.

reduction in all provinces or municipalities from 2020 to 2040. However, in 2050, with increasing NG installed capacity and decreasing CG installed capacity, NGs in Jiangsu, Zhejiang, Hubei, and Shanghai account for a higher compensation proportion. It should be noted that the PV generation reduction in Zhejiang in the different years is similar to that in Jiangsu and even higher than that in Anhui. However, thanks to the higher compensation share of NGs, the total ICEs summed over the different years are lower than those in Jiangsu and Anhui.

**Several pathways to offset the ICEs caused by plum rain**. Plum rain can cause the ICEs of power grids by reducing PV power generation and increasing the output of CGs and NGs. To offset the ICEs, we can adopt measures to prevent increasing the output of CGs and NGs or reduce the CO$_2$ emissions of CGs and NGs when compensating for the missing PV generation attributed to plum rain. First, we consider coal power to natural gas power (C2N) power conversion, i.e., increasing the output of NGs and reducing the output of CGs to ensure that the CO$_2$ emissions of power systems reach a level that is not affected by plum rain. On the basis of C2N power conversion, three promising technology options, including the demand response (DR) program, carbon capture, utilization and storage (CCUS), and long-duration (LD) storage, are considered. An incentive-based DR program, which can offer payments to users to reduce their electricity usage during periods of increased system need or stress in the long term[23], is adopted in the optimization model. Since CGs exhibit a higher carbon emission factor than that of NGs, CCUS is applied to a number of CGs to reduce the CO$_2$ emissions of CGs at the source. LD storage can help shift energy during multiday periods of supply and demand imbalance and thus can be used to store/ release electricity before/during the plum rain period. Here, hydrogen storage is selected as an up-and-coming technology where the energy storage capacity can be designed fully independent of the power capacity[24,25]. Green hydrogen, which is produced by surplus electric power originating from undispatchable renewables, is to be stored via LD storage before the rainy season.

The levelized cost of CO$_2$ mitigation (LCCM)[26], defined as an evaluation index of the additional cost associated with a reduction in CO$_2$ emissions of 1 kg (please refer to the Methods section for details), is adopted to evaluate the performance levels of the C2N, C2N + DR, C2N + CCUS, and C2N + LD pathways. Choosing

Zhejiang as an example, Fig. 5 shows the LCCM and corresponding compensation energy under the several pathways considering different techno-economic parameters. The compensation energy represents the compensation energy of the DR, the clean energy produced by CGs under CCUS, and the net released energy of LD storage excluding the charged energy during the plum rain period. The hollow circle on the border in Fig. 5a–c indicates the LCCM when implementing C2N power conversion alone, which reaches 0.26 ¥/kg in Zhejiang, suggesting that an additional 0.26 ¥ is needed to convert coal power into gas power to reduce 1 kg of CO$_2$ emissions caused by plum rain. Under the C2N + DR pathway, we considered the impact of the DR compensation cost and DR power (measured as percentages of the maximum load during the plum rain period) on the LCCM. As shown in Fig. 5a, the LCCM is mainly affected by the DR compensation cost. As the DR compensation cost decreases from 1.25 to 0.25 ¥/kWh, the LCCM gradually decreases from 0.26 to −0.40 ¥/kg. This indicates that the C2N + DR pathway can reduce system operating costs while reducing CO$_2$ emissions, but double dividends must be realized at a very low DR compensation cost (below 0.50 ¥/kWh). However, the current DR compensation cost in China is relatively high. For example, the current DR compensation price in Zhejiang ranges from a low electricity price of 1.3 ¥/kWh to a peak electricity price of 4 ¥/kWh[27], with a considerable gap at this cost level. Here, we do not show the LCCM at a compensation cost level close to 4 ¥/kWh in Fig. 5a because the compensation energy level is 0 kWh at this current high-cost level with no effects on the LCCM. In contrast, the DR power imposes little effect on the LCCM, and we can see that there is only a limited effect within the range from 2 to 4% of the maximum load. This demonstrates that when the DR technique is adopted to offset the ICEs caused by plum rain, the grid company should seek participants who can actively participate under lower DR cost instead of more participants. Compared to the C2N + DR pathway, the C2N + CCUS pathway exhibits a much smaller optimization space for the LCCM. As the CCUS efficiency changes from 80 to 100% (all CO$_2$ produced by CGs can be captured) and the CCUS cost decreases from 0.5 to 0.1 ¥/kWh, the corresponding LCCM is only reduced from 0.26 to 0.24 ¥/kg. The above results are attributed to the fact that while CCUS effectively captures the CO$_2$ emissions of CGs, this method consumes a considerable amount of electricity (approximately 30%[28]), thereby reducing the efficiency of CGs. Therefore, the use of the C2N + CCUS pathway to offset ICEs does not attain

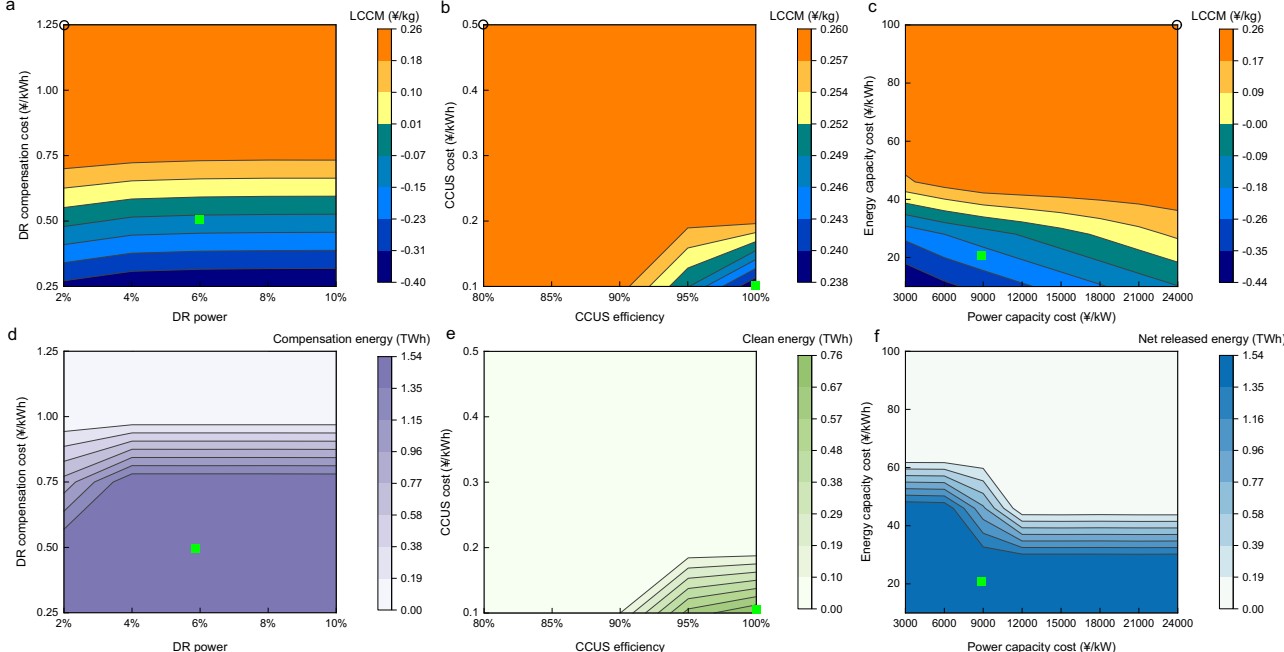

**Fig. 5 Levelized cost of CO$_2$ mitigation (LCCM) and compensation energy under different techno-economic parameters. a** LCCM of coal power to natural gas power (C2N)+ demand response (DR) under different DR compensation cost and DR power levels. **b** LCCM of C2N + carbon capture, utilization, and storage (CCUS) under different CCUS costs and efficiencies. **c** LCCM of C2N + long-duration (LD) under different power and energy capacity cost levels. **d** Compensation energy for C2N + DR under different DR compensation cost and DR power levels. **e** Clean energy for C2N + CCUS under different CCUS costs and efficiencies. **f** Net released energy for C2N + LD under different power and energy capacity cost levels.

competitiveness nor a development potential. Under the C2N + LD pathway, it is found that when the energy capacity cost is higher than 50 ¥/kWh, the change in the power capacity cost exerts little impact on the LCCM. When the energy cost is lower than 50 ¥/kWh, reducing both the power and energy costs can notably reduce the LCCM. The LCCM is even negative for an energy capacity cost of 20 ¥/kWh and a power capacity cost of 21,000 ¥/kW, and the LCCM can reach −0.44 ¥/kg for an energy capacity cost of 10 ¥/kWh and a power capacity cost of 3000 ¥/ kW. The C2N + LD option is a promising pathway, considering that relying on underground storage technology such as salt caverns can satisfy the energy capacity cost requirements at present[29]. However, the power capacity cost, including that of expensive electrolysers and fuel cells, both of which attain the lowest cost higher than 3500 ¥/kW[30,31], cannot be reduced to a low price level in the short term.

Figure 5d–f shows the compensation energy of the DR, clean energy produced by CGs under CCUS, and net energy released by LD storage. As the DR compensation cost declines from 1.25 to 0.25 ¥/kWh and the DR power increases from 2 to 10% of the maximum load, the compensation energy of the DR increases from 0 to 1.54 TWh, and its trend is consistent with that of the LCCM but not synchronized. Compared to the rapid decrease in the LCCM below a DR compensation cost of 0.50 ¥/kWh, the compensation energy increases rapidly when the DR compensation cost varies between 1 and 0.75 ¥/kWh, while it changes slightly when the DR compensation cost is below 0.50 ¥/kWh. CGs with installed CCUS can produce 0.76 TWh of clean electricity at a CCUS cost and efficiency of 0.10 ¥/kWh and 100%, respectively, which exhibits the same change trend as that of the LCCM under the other parameter values. LD storage can release 1.54 TWh of electric energy given the above minimum energy and power capacity costs. In addition, when the energy and power capacity costs range from 30 to 60 ¥/kWh and 6000 to 12,000 ¥/kW, respectively, the net energy released by LD storage

changes sharply, and its change trend is also out of sync with that of the LCCM, which sharply fluctuates when the energy capacity cost is below 40 ¥/kWh. In the other provinces, the first adoption of C2N power conversion results in major differences in the provincial LCCM, among which the highest value is 0.42 ¥/kg in Hunan and the lowest value is 0.12 ¥/kg in Shanghai. The above also leads to subsequent different LCCM decline curves for the various provinces under the different pathways. The C2N + CCUS pathway is only applicable to certain provinces and is very limited in reducing the LCCM. The C2N + DR and C2N + LD pathways can be adopted in all provinces, but the different provinces prefer a certain technology. Details of the LCCM and corresponding compensation energy under the different pathways in the other provinces or municipalities are shown in Supplementary Figs. 3 and 4.

To further reveal the optimal system scheduling results under the different pathways, we also select Zhejiang as an example and determine the hourly power balance during a 1-week period of 168 h from July 2 to 8 in 2040. For display convenience, we select pathways with various LCCMs to clearly show how the different technologies function. The techno-economic parameters corresponding to each technology are shown with a green square mark in Fig. 5a–c. For example, the DR compensation cost and DR power corresponding to the green square in Fig. 5a are 0.50 ¥/kWh and 6%, respectively. Figure 6a shows the power balance of the power system without any measures to offset the plum rain impact, and Fig. 6b–e shows the power balance when the C2N, C2N + DR, C2N + CCUS, and C2N + LD pathways are adopted. Since we focus on how the CG, NG, DR, CCUS, and LD storage technology options operate during this period, the power output, including that of renewables, tie-lines, electric storage, and load curtailment, is summed and denoted as other power. The electric load is also indicated with a black dotted line. In contrast to Fig. 6a, the output of CGs in Fig. 6b decreases, while that of NGs increases, and the corresponding LCCM is 0.26 ¥/kg. On the basis of C2N power

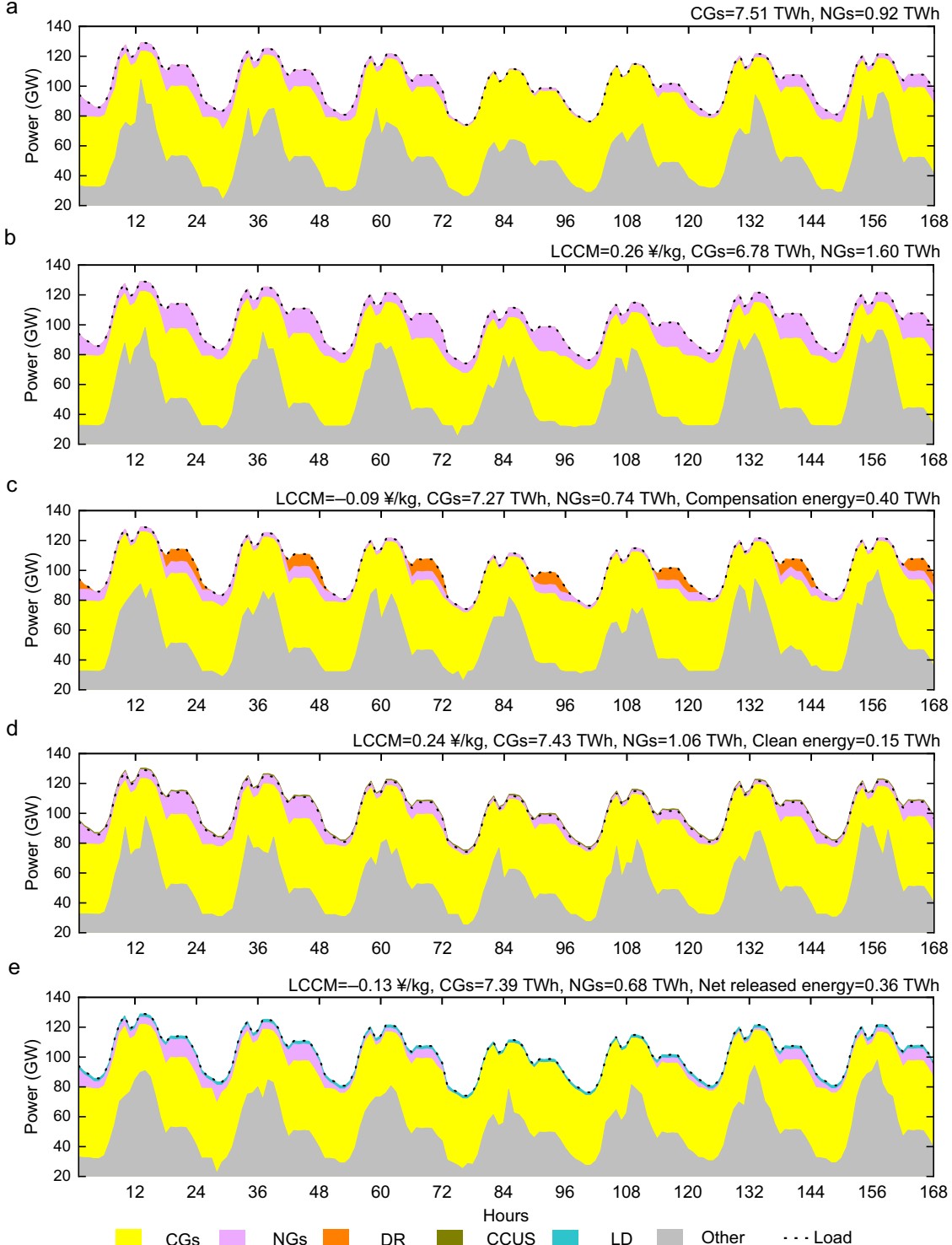

**Fig. 6 Hourly power balance. a** Power balance without any measures. **b** Power balance considering coal power to natural gas power (C2N). **c** Power balance considering C2N + demand response (DR). **d** Power balance considering C2N + carbon capture, utilization and storage (CCUS). **e** Power balance considering C2N + long-duration (LD). Note that "LCCM", "CGs", and "NGs" are short for levelized cost of CO$_2$ mitigation, coal generators, and natural gas generators.

conversion, a DR cost of 0.50 ¥/kWh with a DR power of 6% of the maximum load can achieve compensation energy of 0.40 TWh during this period. Moreover, this reduces gas-fired power generation by 0.86 TWh with a higher fuel cost and increases coal-fired power generation by 0.49 TWh. When replacing the DR option by the CG with the CCUS option, a clean energy amount of

0.15 TWh is produced by CGs with installed CCUS, gas-fired power generation is reduced and coal-fired power generation is increased over C2N power conversion. This holds true for the C2N + LD pathway, except that the net energy released by LD storage reaches 0.36 TWh, and the high power energy cost results in a uniform discharge during this period. The above results

indicate that when no additional technology is implemented, C2N power conversion is an effective means to reduce carbon emissions at the expense of certain economies. On this basis, the DR, CG, CCUS, and LD storage technology options can provide additional clean energy or reduce the demand during the rainy season, and these options require less transfer power than does C2N power conversion alone. Among them, relying on a future reduction in the DR compensation cost and power capacity cost of the LD, C2N + DR, and C2N + LD options are two pathways expected to achieve carbon reduction and economic dividends.

## Discussion

The main purpose of this study was to assess the potential effect of plum rain on the ICEs of power systems in the Yangtze–Huaihe River basin of China. We collected 41-year (1980–2020) hourly SI data from NASA Modern-Era Retrospective Analysis for Research and Applications (MERRA)-2 to measure the impact degree of plum rain on PV generation and the corresponding $CO_2$ emissions of power systems.

The analysis reveals several key results, which could have essential implications for the development of PV systems in the affected region. First, we employ meteorological data to quantify the impact degree on the SI in the affected region during the plum rain period. The impact degree varies greatly in R1 (covers nearly 500,000 km$^2$). Among the areas, the impact degree in the intersection of Shanghai, Jiangsu, and Zhejiang can reach 21%, which could reduce the electric potential of the power system with high PV integration. Second, plum rain can already cause ICEs of 1.22 megatons per year at the current PV deployment scale. With increasing PV capacity in the future, ICEs could continue to grow and reach a peak value of 5.16 megatons in 2040 in the base case. In addition, different generation and demand change trends exert an important impact on ICEs. Third, the ICEs can be offset by converting coal power into natural gas power. However, the LCCM varies in the different provinces, ranging from 0.12 to 0.42 ¥/kg. Considering a DR compensation cost below 0.5 ¥/kWh or an energy capacity cost of 10 ¥/kWh using underground hydrogen storage technology, coupled with C2N power conversion, all provinces can achieve carbon reduction and economic dividends.

Despite these findings, there are a number of limitations that should be acknowledged. First, although we applied a certain change trend interval to characterize future generation and demand growth, the change trends in certain provinces may deviate from this interval and thus alter the total ICEs to a certain extent. Second, to more notably reduce uncertainties in the CUC model, we did not consider the development trend of other renewable energy sources in the provincial power systems. However, the above assumptions do not affect the following: when plum rain causes PV generation reduction, easily dispatchable CGs and NGs are always the most convenient way to fill the gap in power generation. Third, due to the lack of network congestion in the provincial CUC model, a high penalty cost results in an extremely small amount of PV curtailment, which to a certain extent overestimates the impact of plum rain on PV generation.

The nature of the effect of plum rain on ICEs involves the increase in fossil energy power generation of the power system. With the gradual elimination of thermal power among power systems, the $CO_2$ emissions in the electricity sector will eventually reach zero. Although many studies have demonstrated that renewable energy sources such as wind and solar energy exhibit a high power generation potential, they are sufficient to meet the energy demands of people[32–35]. However, achieving the maximum utilization of deployed renewables remains a goal to be pursued. Therefore, to improve the level of renewable energy utilization, on the one hand, the level of power generation should be enhanced, such as conversion efficiency improvement, and on the other hand, the negative impact of renewable energy utilization on the energy system should be reduced, which is also the focus of this paper. In China and other parts of the world, the development of renewables is influenced by climate conditions[18,20]. If the layout can be designed in advance according to regional climate conditions, wind and solar power benefits could be maximized, which plays a positive role in ensuring regional energy security and stability.

## Methods

**Space–time boundary of plum rain**. To effectively measure the impact degree of plum rain on SI and PV generation, we first clearly define the space–time boundary of plum rain. According to the national standard (GB/T 33671-2017) of plum rain monitoring indices implemented on 12/01/2017, the plum rain-affected region covers the Yangtze–Huaihe River basin, including Jianghuai District, the Yangtze River Middle, and Lower Reaches and Jiangnan District, with an affected region of nearly 500,000 km$^2$ [36]. On this basis, we further extend the affected region by 200 km, obtaining a surrounding contrast region (R2) with a similar size. The average SI value in R2 is chosen as a criterion for the region not affected by plum rain. Under normal circumstances, the duration of the plum rain period extends from mid-June to mid-July of each year. Here, we select 35-day data from the 26th to the 29th week of the year, and the corresponding period lasts from June 18 to July 22 (it is assumed that the first day of the year is January 1, which is also the first day of the first week). This duration time can characterize the process of the gradual strengthening and fading of the impact degree of plum rain.

**SI data and CF calculation**. A dataset containing 41-year (1980–2020) hourly RSISF data with a spatial resolution of 1/2° latitude by 2/3° longitude, retrieved from MERRA-2[37], is employed to measure the impact degree of plum rain on the SI in R1. The 41-year SI value is averaged to 1 year with 8760 values at each spatial resolution. Considering that week-level data can better reveal the impact of plum rain than can day-level (high volatility) and month-level (insufficient accuracy) data, we further average the 8760 values into 52 week-level values. On this basis, the average SI in the region (R1 or R2) or province/municipality can be calculated by further averaging the SI at all spatial resolutions within the region or province/municipality. The impact degree of plum rain on the SI at a given location in R1 can be calculated as the ratio of the difference between $I_{SI}^{MEAN}(w)$ and $I_{SI}(c, w)$ to $I_{SI}^{MEAN}(w)$, where $I_{SI}^{MEAN}(w)$ is the mean SI during the $w$th week in R2, and $I_{SI}(c, w)$ is the SI at a given location $c$ in R1. $M_{R1}$ denotes the set of spatial resolutions in R1, and $\Phi^W$ is the set of weeks during the plum rain period.

$$\theta^{MEAN}(c, w) = \frac{I_{SI}^{MEAN}(w) - I_{SI}(c, w)}{I_{SI}^{MEAN}(w)}, \forall c \in M_{R1}, w \in \Phi^W \quad (1)$$

The open-source Global Solar Energy Estimator (GSEE) model[38] available on the www.renewables.ninja web platform is adopted to estimate PV CFs, which are applied in the CUC optimization model. With the use of meteorological data acquired from MERRA-2, the model considers the inputs of both the instantaneous irradiance and temperature and can regulate the solar plane tilt angle at a given location to yield better outputs. Considering that the plum rain-affected areas occur near latitude 30 °N, we set the optimum tilt angle of the fixed-tilt system to 26.6 degrees, following the study by Chen et al.[32]. In the provinces in both R1 and R2, we consider the locations in each province with a spatial resolution of 1/2° latitude by 2/3° longitude and calculate the mean unit PV output in R1 and R2, respectively (Supplementary Fig. 5). The calculated hourly mean unit PV outputs are considered for future projections of the PV potential. Additionally, a detailed description of the GSEE model can be found in Supplementary Note 1.

**Electric power system data**. Coal-fired generation data, gas-fired generation data, PV generation data, wind power generation data, other non-fossil energy power generation data, and line transmission data for Jiangsu, Anhui, Zhejiang, Jiangxi, Hubei, Shanghai, and Hunan are collected from the China Electric Power Statistical Yearbook[39], China Power Knowledge[40], and power grid companies. Apart from the PV generation capacity in the different provinces or municipalities, at the end of December 2020, the total wind generation capacity in Jiangsu, Anhui, Zhejiang, Jiangxi, Hubei, Shanghai, and Hunan reached 15.47, 4.12, 1.86, 5.10, 5.02, 0.82, and 6.69 GW, respectively, the total hydropower generation capacity in Jiangsu, Anhui, Zhejiang, Jiangxi, Hubei, Shanghai, and Hunan was 2.65, 4.74, 11.71, 6.60, 37.57, 0, 15.81 GW, respectively, the total nuclear power generation capacity in Jiangsu and Zhejiang was 5.49 and 9.11 GW, respectively, and that in the other provinces reached 0 GW. The actual output of hydropower and nuclear power is calculated based on the annual utilization hours and installed capacity of generators. According to the 2020 provincial statistical yearbooks, the annual interactive electricity in Jiangsu, Anhui, Zhejiang, Jiangxi, Hubei, Shanghai, and Hunan reached 120.1, −58.6, 135.52, 15.98, −84.0, 86.96, and 69.25 TWh, respectively.

According to the China Electric Power Statistical Yearbook 2020, the total thermal generation capacity in Jiangsu, Anhui, Zhejiang, Jiangxi, Hubei, Shanghai,

and Hunan is 100.79, 55.61, 63.58, 24.55, 33.16, 24.50, and 22.69 GW, respectively. The coal-fired generation capacity in each province can be obtained from the Global Coal Plant Tracker[41]. Then, the gas-fired generation capacity can be obtained by subtracting the coal-fired generation capacity from the thermal generation capacity. Considering that the NG capacity largely remains below 600 MW, we divide the thermal power generating units below 600 MW into CGs and NGs based on the ratio of the coal-fired generation capacity to the gas-fired generation capacity.

Due to data limitations, the hourly provincial electric loads from 6/18/2020 to 7/22/2020 are collected from provincial power grid companies or fitted based on typical provincial daily and annual load curves. In the base case, we consider that the capacity change trends of coal-fired generators, gas-fired generators, PV systems, interchange tie-lines, and electricity loads over time in the different provinces or municipalities are the same as the relevant predicted change trends in China referenced from the China Energy and Electricity Outlook[22].

**Electricity dispatch model.** A CUC model[42] is employed in each province to calculate the $CO_2$ emissions of CGs and NGs in the different cases, with the advantages of the use of easily available clustered unit data as input to improve the computational efficiency[34]. The objective function in Eq. (2) minimizes the total cost of province-level power systems during the plum rain period, comprising the generation cost of thermal generators (TGs, including CGs and NGs) in the first row, the penalty costs for solar curtailment in R1 (the second row) and R2 (the third row), and the additional storage cost for PV integration and the penalty cost for load curtailment in the fourth row.

$$\min_{X} \sum_{i \in \Phi^{TG}} \sum_{t \in \Phi^T} [c_i^{TG} P_i^{TG}(t) + c_i^U Y_i(t) + c_i^D Z_i(t)]$$
$$+ \varphi^{PV} \sum_{l_1 \in \Phi^{PV}_{R1}} \sum_{t \in \Phi^T} \left[ \rho^{PV}_{R1}(t) C^{PV}_{l_1} - P^{PV}_{l_1}(t) \right]$$
$$+ \varphi^{PV} \sum_{l_2 \in \Phi^{PV}_{R2}} \sum_{t \in \Phi^T} \left[ \rho^{PV}_{R2}(t) C^{PV}_{l_2} - P^{PV}_{l_2}(t) \right] \quad (2)$$
$$+ c_{PR}^{ES} C^{ES} + \varphi^{Loss} \sum_{t \in \Phi^T} P^{Loss}(t)$$

Subject to

$$\sum_{i \in \Phi^{TG}} P_i^{TG}(t) + \sum_{k \in \Phi^{WT}} P_k^{WT}(t) + \sum_{l_1 \in \Phi^{PV}_{R1}} P^{PV}_{l_1}(t) + \sum_{l_2 \in \Phi^{PV}_{R2}} P^{PV}_{l_2}(t)$$
$$+ \sum_{m \in \Phi^{RG}} P_m^{RG}(t) + \sum_{n \in \Phi^{TL}} P_n^{TL}(t) + P^{ES-}(t) - P^{ES+}(t) = P^{Load}(t) - P^{Loss}(t), \forall t \in \Phi^T \quad (3)$$

$$\sum_{i \in \Phi^{TG}} P^{TG}_{i,\max} U_i(t) + \sum_{m \in \Phi^{RG}} P_m^{RG}(t) + \sum_{n \in \Phi^{TL}} P_n^{TL}(t) + S^{ES}(t-1) \geq P^{Load}(t) + RS(t), \forall t \in \Phi^T \quad (4)$$

$$P^{TG}_{i,\min} U_i(t) \leq P_i^{TG}(t) \leq P^{TG}_{i,\max} U_i(t), \forall i \in \Phi^{TG}, t \in \Phi^T \quad (5)$$

$$Y_i(t) - Z_i(t) = U_i(t) - U_i(t-1), \forall i \in \Phi^{TG}, t \in \Phi^T \quad (6)$$

$$U_i(t) \geq \sum_{\delta=t-T_i^U+1}^{t} Y_i(\delta), \forall i \in \Phi^{TG}, t \in \Phi^T \quad (7)$$

$$N_i - U_i(t) \geq \sum_{\delta=t-T_i^D+1}^{t} Z_i(\delta), \forall i \in \Phi^{TG}, t \in \Phi^T \quad (8)$$

$$u_i^{g+1}(t) \leq u_i^g(t), \forall i \in \Phi^{TG}, \forall g \in [1, N_i), t \in \Phi^T \quad (9)$$

$$u_i^1(t) \leq 1, u_i^{N_i}(t) \geq 0, \forall i \in \Phi^{TG}, t \in \Phi^T \quad (10)$$

$$0 \leq P^{PV}_{l_1}(t) \leq (1 + \gamma^{aerosol}) \rho^{PV}_{R1}(t) C^{PV}_{l_1}, \forall l_1 \in \Phi^{PV}_{R1}, t \in \Phi^T \quad (11)$$

$$0 \leq P^{PV}_{l_2}(t) \leq (1 + \gamma^{aerosol}) \rho^{PV}_{R2}(t) C^{PV}_{l_2}, \forall l_2 \in \Phi^{PV}_{R2}, t \in \Phi^T \quad (12)$$

$$0 \leq P^{ES+}(t) \leq \frac{C^{ES}}{4}, \forall t \in \Phi^T \quad (13)$$

$$0 \leq P^{ES-}(t) \leq \frac{C^{ES}}{4}, \forall t \in \Phi^T \quad (14)$$

$$S^{ES}(t) = S^{ES}(t-1) + \eta^{ES+} P^{ES+}(t) - \frac{P^{ES-}(t)}{\eta^{ES-}}, \forall t \in \Phi^T \quad (15)$$

$$0 \leq S^{ES}(t) \leq C^{ES}, S^{ES}(0) = S^{ES}(T) = \frac{C^{ES}}{2}, \forall t \in \Phi^T \quad (16)$$

$$U_i(t), Y_i(t), Z_i(t) \in \{0, 1, \cdots, N_i\}, \forall i \in \Phi^{TG}, t \in \Phi^T \quad (17)$$

where the decision variable set $X$ includes the hourly power of the $i$th clustered type

of TGs $P_i^{TG}(t)$, the hourly PV outputs of the $l_1$th PV generator $P^{PV}_{l_1}(t)$ in R1 and the $l_2$th PV generator $P^{PV}_{l_2}(t)$ in R2, the hourly charging/discharging power $P^{ES+}(t)/P^{ES-}(t)$, the hourly electric load curtailment $P^{Loss}(t)$, the maximum power capacity of energy storage $C^{ES}$, an integer variable indicating the number of operating $i$th clustered types of TGs $U_i(t)$, and an integer variable indicating the number of $i$th clustered types of TGs in the startup/shutdown state $Y_i(t)/Z_i(t)$. Except for the decision variables in the objective function (Eq. (2)), $\rho^{PV}_{R1}(t)/\rho^{PV}_{R2}(t)$ is the unit mean PV output in R1/R2, $C^{PV}_{l_1}/C^{PV}_{l_2}$ is the installed capacity of the $l_1$th PV generator in R1 or the $l_2$th PV generator in R2, $c_i^{TG}$ is the cost per MWh of the $i$th clustered type of TG, $c_i^U$ and $c_i^D$ are the corresponding startup and shutdown costs, respectively. In addition, $\varphi^{PV/Loss}$ in the objective function (Eq. (2)) is the penalty factor in regard to solar power/load curtailment, and a sufficiently large value of 100 ¥/kWh is set to avoid solar power and load curtailments as much as possible. $c_{PR}^{ES}$, which is equal to $c^{ES} \cdot \frac{T}{8760} \cdot \frac{r(1+r)^{n^{ES}}}{(1+r)^{n^{ES}}-1}$, is the unit investment cost of energy storage during the plum rain period. In this equation, $c^{ES}$ and $n^{ES}$ are the unit investment cost and lifetime, respectively, of electric storage, $T$ defines the number of hours during the plum rain period, and $r$ is the discount rate. $\Phi^{TG}$ is the set of TGs, and $\Phi^{PV}_{R1}$ and $\Phi^{PV}_{R2}$ are the sets of PV generators in R1 and R2, respectively. $\Phi^T$ is the set of hours during the plum rain period.

Equation (3) expresses the balance for the power supply and load demand, where $P_k^{WT}(t)$, $P_m^{RG}(t)$, and $P_n^{TL}(t)$ are the hourly power levels of the $k$th wind power plant, the $m$th renewable generator (comprising hydroelectric and nuclear power units), and the $n$th interchange tie-line, respectively, and $P^{Load}(t)$ is the hourly total electric load. $\Phi^{WT}$, $\Phi^{RG}$, and $\Phi^{TL}$ denote the sets of wind plants, renewable power generators, and interchange tie-lines, respectively. Constraint (4) expresses the system reserve requirements, where $P^{TG}_{i,\max}$ is the unit mean capacity of the $i$th clustered type of TG, $S^{ES}(t-1)$ is the ending hour-$t$-1 state of charge (SOC) of electric storage, and $RS(t)$ is the hourly upward reserve margin[43]. Constraint (5) bounds the lower and upper limits of the $i$th clustered type of TG, where $P^{TG}_{i,\min}$ is the minimal power for one unit when the $i$th clustered type of TG is brought online. Constraints (6)–(8) express the relationships between $U_i(t)$, $Y_i(t)$, and $Z_i(t)$, and limit the minimum on/off hours of the $i$th clustered type of TG, where $T_i^U$ and $T_i^D$ are the minimum on and off hours, respectively, of the $i$th clustered type of TG, and $N_i$ is the total number units in the $i$th clustered type of TG[44]. Constraints (9)–(10) impose the commitment order that TG 1 is committed first and subsequently TG $N_i$ is committed last, where $u_i^g(t)$ is a binary variable indicating the on/off state of TG $g$ in the $i$th clustered type of TG[46]. Based on similar nameplate capacities, six CG groups are clustered, and four NG groups are clustered (please refer to Supplementary Table 3 for details). Constraints (11) and (12) define the lower and upper limits of PV power generation in R1 and R2, respectively, where $\gamma^{aerosol}$ is the percentage increase in the PV potential caused by anthropogenic aerosol emissions. Following the study of Sweerts et al.[13], we assume that Chinese anthropogenic aerosol emissions in 2060 are consistent with those in 1960, which could yield a 12–13% increase in PV generation by improving the all-sky SI. Although the $CO_2$ emissions in China will continue to increase and a peak is expected to occur in 2030, the implementation of ultralow emissions standards for coal-fired power plants[45] and alternate electric power policies such as electric heating policies[44] have improved the clean use of coal. Therefore, starting in 2020, we consider the PV potential to increase by 3% every 10 years. Constraints (13)–(15) impose power capacity and energy constraints on the charging, discharging, and SOC levels of energy storage, where $\eta^{ES+/-}$ is the charging/discharging efficiency of energy storage. Constraint (16) ensures a nonnegative SOC value and that the starting and ending states remain the same. Here, Li-ion storage with a duration up to 4 h at the rated power[25] is chosen as a representative option, and the energy capacity cost is set to 1380 ¥/kWh[47]. Constraint (17) defines the bounds of the integer variables.

The $CO_2$ emissions of TGs are calculated as follows:

$$E = e^C \sum_{p \in \Phi^{TG}_C} \sum_{t \in \Phi^T} P_p^{TG}(t) + e^N \sum_{q \in \Phi^{TG}_N} \sum_{t \in \Phi^T} P_q^{TG}(t) \quad (18)$$

where $e^C$ and $e^N$ are the emission factors of the CGs and NGs, respectively, $\Phi^{TG}_C$ and $\Phi^{TG}_N$ denote the sets of CGs and NGs, respectively, $P_p^{TG}(t)$ is the hourly power of the $p$th clustered type of CG, and $P_q^{TG}(t)$ is the hourly power of the $q$th clustered type of NG. The sum of the two amounts equals the outputs of the TGs at any time. The fuel costs and emission factors of CGs and NGs are summarized in Supplementary Table 4.

By replacing constraint (11) with constraint (19) and re-optimizing the CUC model, the power system operation status and corresponding carbon emissions eliminating any plum rain effects can be obtained.

$$0 \leq P^{PV}_{l_1}(t) \leq \left( \frac{1}{1 - \theta_P^{MEAN}(w)} \right) \cdot (1 + \gamma^{aerosol}) \rho^{PV}_{R1}(t) C^{PV}_{l_1}, \forall l_1 \in \Phi^{PV}_{R1}, w \in \Phi^W, t \in \Phi^T_w \quad (19)$$

Where $\theta_P^{MEAN}(w)$ is the provincial mean impact degree during the $w$th week and $\Phi^T_w$ is the set of hours during the $w$th week.

Finally, the ICEs $\Delta E$ caused by plum rain can be obtained as the difference between the $CO_2$ emissions determined via Eq. (18) under the first optimization $E^{(1)}$ and those determined under the second optimization $E^{(2)}$.

$$\Delta E = E^{(1)} - E^{(2)} \tag{20}$$

**Comparison of the various technologies**. According to the different mitigation principles of carbon emissions, we consider four pathways to address the negative impacts of plum rain: C2N, C2N + DR, C2N + CG with CCUS, and C2N + LD. The CUC models considering C2N, C2N + DR, C2N + CG with CCUS, and C2N + LD are presented in Supplementary Note 2. To compare their performance levels, we further define the LCCM as follows:

$$\text{LCCM} = \frac{\Delta C}{\Delta E} \tag{21}$$

where $\Delta C$ is the total additional cost when offsetting the ICEs caused by plum rain.

**Reporting summary**. Further information on research design is available in the Nature Research Reporting Summary linked to this article.

## Data availability
The 41-year hourly radiation-surface incoming shortwave flux (RSISF) data from 1980 to 2020 with a spatial resolution of 1/2° latitude by 2/3° longitude are available from the MERRA-2 (https://disc.gsfc.nasa.gov/datasets/M2T1NXRAD_5.12.4/summary?keywords=incident%20shortwave%20radiation%20flux). Unit-level data of provincial coal-fired power plants used in this analysis are from the Global Coal Plant Tracker (https://endcoal.org/globalcoal-plant-tracker/). All data to generate Figs. 1–6 and Supplementary Figs. 3 and 4 are available in the Supplementary Data file.

## Code availability
The code used in this study is available from the authors upon request.

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

## Acknowledgements
We acknowledge the support from the National Natural Science Foundation of China (U1866208), the Natural Science Foundation of Jiangsu Province (Grant no. BK20200013), the National Key R & D Program of China (2020YFE0200400), and the Scientific Research Foundation of Graduate School of Southeast University (YBPY2036).

## Author contributions

G.P. and Q.H. conceived and designed the research. G.P., Q.H., W.G., and Y.L. developed the framework and formulated the theoretical model. G.P., Q.H., S.D., and H.Q. carried out the data search. G.P., S.D., and H.Q. carried out the simulations and analyses. All authors contributed to the discussions on the method and the writing of this article.

## Competing interests

The authors declare no competing interests.
