## [Peer Review File · Nature Communications]

Reviewer #3 (Remarks to the Author):

This paper discusses the role of "plum rain" on solar power potential in China and potential solutions to balance electricity generation during these times. The idea of this paper, on its surface, is interesting and highly topical as countries gear towards more decarbonization commitments. The authors clearly committed significant work to this project, but there are a number of details missing that I think, at present, make the paper unfit for acceptance. With significant improvements to the manuscript, I think that this study could become more appropriate for Nature Communications.

Major comments:

1) Solar PV: I'm a bit confused as to how solar power potential was calculated in this paper. The authors reference Chen et al. (2019), but it appears that the temperature dependence of solar power electricity generation is not included in the calculations done here (which should be an important consideration in a paper that discusses the effects of climate change on solar power potential). Furthermore, more information is needed as to what solar power potential data are used for the Unit Commitment model. Are future projections of solar potential based on the average meteorological conditions from 2000-2019? I think that's reasonable, but should be justified in the text. Also, some of the analysis studies the data at the city-level, but isn't the reanalysis product too spatially coarse to analyze at such a granular level?

2) Unit commitment model assumptions: Implications of potential plum rain effects on the the electricity grid are certainly interesting, but I think there is a lot of missing information for how the model is run. For instance, assumptions of cost for various technologies are one of the most important components of a unit commitment model. As far as I can tell, there is no information on costs in the text (except for a brief mention of three cost scenarios), nor is there any justification for the costs assumed. This should, at the very least, be a significant component of the methods section. Technologies like wind and nuclear are included in the methods section, but are not mentioned in the results – are these included in the unit commitment modeling?

3) Mitigation methods: Similarly to my prior comments, I think there needs to be many more details for this section. What's the difference between Hydrogen1 and Hydrogen2 (if any at all besides discharge time)? What's the source of the hydrogen; is this grey, blue or green hydrogen? What are potential methods of demand response changes? It would also help to clarify the captions for figures 3 and 4 which I found to be confusing.

Minor comments:

L12: Should explain what "plum rain" is in the abstract.

L51: I think this is meant to be "2060" instead of "3060".

L82-83: I think this discussion of climate change is confusing because it's not actually being investigated in the paper.

L108-109: This should be elaborated further here as to why that's the case (it makes sense after reading the rest of the paper, but it was unclear at the time of reading).

L130: The trend doesn't seem statistically significant

L130-132: How can you conclude that 2020 has a $\Delta\theta$ above 7%? Is this not shown in the figure?

L144-145: The caption should include the definition of $\Delta\theta_{\text{TIME}}$.

Figure 1, panel c: It would be helpful to identify R1 and R2 in the figure so the reader doesn't have to look through the caption to find what each is referring to.

L168: What are the numbers for PV capacity quoted here? Do these represent the projected 2050 PV capacity or something else?

L190-191: How can this be concluded? The climate data used in the paper goes from 2000-2019, so how can conclusions be made about the PV output from 2020-2050? Is there information missing?

L214-215: Should include discussion of potential methods for reducing power load.

L216: Capacity is units for GW, not GWh. Also, what does it mean for DR energy capacity to reach

2900 GWh? Is that saying how much electricity you would need to save by reducing power load?

Table 1: Missing "1" and "2" for hydrogen1 and hydrogen2 in the table. Though I'm still not sure what the difference is between the two and why it's labelled as such.

Figure 3: I found this figure highly confusing. Does each dot represent a scenario run in the unit commitment model? What do the units of GWh mean for DR and CCUS?

L275-276: There needs to be discussion of how the prices assumed for the paper.

Figure 4: There seems to be a lot of missing information for this figure. What does each of these subplots show and why is the y axis different for each technology studied? Also, what does the time on the x axis mean in the context of this figure?

L345: this should say "...CO2 emissions from the electricity sector...", specifically.

L349-350: I think this point about renewables taking up land is a bit of an overstatement, especially for China where the optimal wind resources are in Inner Mongolia and offshore.

Reply to the comments

Manuscript ID: NCOMMS-21-16660A

Title: Assessment of the plum rain impact on power system emissions in the Yangtze-Huaihe River basin of China

Dear Reviewers:

We would like to thank the Reviewers for the insightful comments and for the thorough revision of the manuscript. All the comments and suggestions have been carefully addressed and **highlighted in red** in the revised manuscript. In what follows, we detail the revisions made in response to the Reviewers' suggestions and concerns. For convenience, the comments and suggestions of the reviewers are **highlighted in blue**, whereas our response and statements are in regular font and start after the word of "Answer".

Reviewer #1 (Remarks to the Author):

The paper deals with a subject that falls within the scope of the journal. Language is adequate, although proof reading by a native speaker could improve it substantially.

The layout of the paper should be reorganized. After the introduction it would be better to provide the methodology, followed by the results and the discussion.

Answer:

We appreciate the Reviewer's invaluable suggestions and comments.

Your positive affirmation of our work prompted us to further improve the quality of this work. The manuscript has been carefully revised according to the Reviewer's comments. The detailed replies are presented one by one as follows.

The manuscript has been carefully re-edited by the Springer Nature Author Services (Order ID CNNDZ1KY) to ensure that the language meets the requirements.

At the end of the introduction, we briefly summarized the methodology in this paper, which can help readers better understand this paper. The specific and complex modeling process is placed in the Methods at the end of the paper, which is also in line with most paper formats published in *Nature communications*.

Furthermore a brief description of the plum rain effect should be provided.

Answer:

We appreciate the Reviewer's invaluable suggestions and comments.

We have added the following description of the plum rain effect in the revised manuscript.

As an important product of the northward advancement of the East Asian summer monsoon, plum rain mainly occurs in the Yangtze-Huaihe River basin of China, affecting an area of nearly 500000 km².^[R1] ***During the plum rain period, continuous cloudy and rainy weather conditions prominently occur, which can easily cause floods, reduce crop yields, and affect people's transportation patterns. Additionally, clouds and precipitation during the plum rain period can reduce the surface irradiance (SI), yielding economic and carbon challenges to the operation of power systems by reducing the PV potential.***

The authors state that 50 representative cities have been considered but they do not state how they were selected or why they are representative. A table in the main text with the location and the main to information regarding climatic data and energy mix for each city would help.

Answer:

We appreciate the Reviewer’s invaluable suggestions and comments.

The selection principle of the initial 50 cities is mainly based on the regional distribution within the affected region. For example, we have selected six cities in the middle, east, west, south, north, and middle in Jiangsu to make the cities evenly distributed in the affected areas as much as possible. The energy mix is only considered at the province-level, not city-level, in the initial manuscript.

In the revised manuscript, we have adopted a more straightforward and more intuitive method to compare the effects of surface radiation in the affected and surrounding regions during the plum rain period through 41-year hourly surface irradiance (SI) with a spatial resolution of 1/2° latitude by 2/3° longitude from 1980 to 2020 derived from NASA’s MERRA-2.^[R2] This method can avoid the one-sided impact of the uneven selection of representative cities on the evaluation of plum rain’s effects; see Supplementary Fig. 5. A compromise between accuracy and ease of analysis and display, a weekly year-average of the above 41-year data is used to evaluate the effects.

Supplementary Figure 5 Spatial resolution of 1/2° latitude by 2/3° longitude to calculate PV capacity factor.

In addition, the area ratio, impact degree, and PV capacity factor (CF) in each affected province/municipality are shown in Supplementary Table 2. The energy mix of each provincial power grids is given in the *Electric power system* data in *Methods*.

Supplementary Table 2. Area ratio, impact degree, and PV capacity factor (CF) in each affected province/municipality

Province	Area ratio	θ_p^{MEAN}	PV capacity factor ^[R3]
Jiangsu	80%	7.2 %	0.17
Anhui	77%	7.0 %	0.17

Zhejiang	70%	10.2%	0.156
Jiangxi	45%	7.8 %	0.156
Hubei	53%	4.3%	0.172
Shanghai	100%	10.2%	0.164
Hunan	18%	5.0 %	0.162

In lines 78-81 more details regarding the references provided should be given. Also the literature review of the specific subject is very cursory discussed and should be thoroughly expanded.

Answer:

We appreciate the Reviewer's invaluable suggestions and comments.

In the revised manuscript, we have given more details regarding the references provided. Also, we have thoroughly expanded the depth of the literature review to highlight the focus and innovation of our work. We have added the following contents in the revised manuscript.

For instance, anthropogenic aerosol emissions and changes in cloud cover^[R4] and frequent extreme conditions^[R5] can reduce the PV potential. The warming of the Indian Ocean can lead to a secular decrease in the wind power potential in India.^[R6] Climate change can decrease the dry season hydropower potential, thus worsening the mismatch between the seasonal electricity supply and peak demand.^[R7] As a result, future energy systems dominated by renewables can face challenges in the reliability of energy supply. Therefore, it is necessary to deeply understand the negative impacts of typical climate caused by potential factors before the large-scale implementation of renewable energy techniques.

The legend in Figures 1 and 2 are too long. It is recommended that the figures are broken in two or more.

Answer:

We appreciate the Reviewer's invaluable suggestions and comments.

As suggested by the reviewer, the legend in Figs. 1 and 2 have been shortened in the revised manuscript. Fig. 1, with three subgraphs in the initial manuscript, has been split into Figs. 1 and 2 in the revised manuscript. Fig. 2, with four subgraphs in the initial manuscript, has been divided into Figs. 3 and 4 in the revised manuscript. Detailed modifications can be seen in the revised manuscript.

In lines 224-235 the authors need to provide the reasoning behind investigating the effect of the specific technologies taken into consideration as representative technologies.

Answer:

We appreciate the Reviewer's invaluable suggestions and comments.

We have added the following contents in the revised manuscript to provide the reasoning behind the investigating the effect of the specific technologies taken into consideration as representative technologies.

Plum rain can cause the ICEs of power grids by reducing PV power generation and increasing the output of dispatchable CGs and NGs. To offset the ICEs, we can adopt measures to prevent increasing the output of CGs and NGs or reduce the CO₂ emissions of CGs and NGs when compensating for the missing PV generation attributed to plum rain. First, we consider CG-to-NG (C2N) power conversion, i.e., increasing the output of NGs and reducing the output

of CGs to ensure that the CO₂ emissions of power systems reach a level that is not affected by plum rain. **The advantage of this pathway is that there is no need to add any additional equipment, and it is implemented directly through optimized scheduling.**

On the basis of C2N, three promising technologies, including demand response (DR), carbon capture, utilization and storage (CCUS), and long-duration (LD) storage, are selected based on the principle of not increasing the output of thermal generators or reducing CO₂ emissions of them. An incentive-based DR program, which can offer payments to users to reduce their electricity usage during periods of the increased system need or stress in the long term,^[R8] is adopted in the optimization model. Since CGs exhibit a higher carbon emission factor than that of NGs, CCUS is applied to a number of CGs to reduce the CO₂ emissions of CGs at the source. LD storage can help shift energy during multiday periods of supply and demand imbalance and thus can be used to store/release electricity before/during the plum rain period. Here, hydrogen storage is selected as an up-and-coming technology where the energy storage capacity can be designed fully independent of the power capacity.^{[R9],[R10]} Green hydrogen, which is produced by surplus electric power originating from undispatchable renewables, is to be stored via LD storage before the rainy season.

The main shortcoming of the paper is that as the authors also state the development trend of the line transmission capacity was not considered, and also that the renewable trend in the provinces follows that of China which might not be the case. A comparison of the historical evolution or RES in the specific provinces and China could help.

Answer:

We appreciate the Reviewer’s invaluable suggestions and comments.

In the revised manuscript, we have added the development trend of the line transmission capacity in the optimization model. Considering that the renewable trend in the provinces follows that of China which might not be the case, **we have used a change trend interval to evaluate the incremental CO₂ emissions in different years.** The change trends of coal power, gas power, photovoltaic, line transmission capacity, and electric load in different years are shown in Supplementary Table 1. Among them, the base case refers to the national average change trends forecasted by the State Grid Energy Research Institute. **In order to cover the difference in the change trends of different provinces, the average change trends are fluctuated up and down by 25% on the base case.**

Supplementary Table 1. Predicted change trends of different generators demand.^[R11]

Type	2020	2030	2040	2050
Coal power	1	1.3	1.09	0.74
Natural gas power	1	1.9	3	3.9
Photovoltaic	1	2.87	4.94	6.56
Interchange tie-line	1	1.4	1.8	2.2
Power load	1	1.49	1.74	1.83

All the simulation results have been accordingly revised.

In the end, the authors would like to thank the Reviewer for all the invaluable suggestions and comments on this paper. It is your kind help that makes our work

better.

References

- R1 China Meteorological Administration. Plum rain monitoring indicators. http://www.cma.gov.cn/kppd/kppdrt/201607/t20160701_315499.html (2016).
- R2 Gelarova, R., McCarty, W., and Suárez, M. J. et al. The modern-era retrospective analysis for research and applications, version 2 (MERRA-2). *J. Clim.* **30**, 5419–5454 (2017).
- R3 Pfenninger, S. & Staffell, I. Long-term patterns of European PV output using 30 years of validated hourly reanalysis and satellite data. *Energy* **114**, 1251–1265 (2016).
- R4 Sweerts, B., Pfenninger, S., Yang, S. et al. Estimation of losses in solar energy production from air pollution in China since 1960 using surface radiation data. *Nat. Energy* **4**, 657–663 (2019).
- R5 Feron, S., Cordero, R.R., Damiani, A. et al. Climate change extremes and photovoltaic power output. *Nat. Sustain.* **4**, 270–276 (2021).
- R6 Gao, M., Ding, Y., Song, S. Lu, X., Chen, X., McElroy, M. B. Secular decrease of wind power potential in India associated with warming in the Indian Ocean. *Sci. Adv.* **4**, eaat5256 (2018).
- R7 Arias, M.E., Farinosi, F., Lee, E. et al. Impacts of climate change and deforestation on hydropower planning in the Brazilian Amazon. *Nat. Sustain.* **3**, 430–436 (2020).
- R8. Parvania, M. and Fotuhi-Firuzabad, M. Demand response scheduling by stochastic SCUC. *IEEE Trans. Smart Grid* **1**, 89–98 (2010).
- R9 Sepulveda, N.A., Jenkins, J.D., Edington, A. et al. The design space for long-duration energy storage in decarbonized power systems. *Nat. Energy* **6**, 506–516 (2021).
- R10 Albertus, P., Manser, J. S., and Litzelman, S. Long-duration electricity storage applications, economics, and technologies. *Joule* **4**, 21–32, (2020).
- R11 State Grid Energy Research Institute. China Energy & Electricity Outlook (China Electric Power Press, 2019).

Reviewer #2 (Remarks to the Author):

Sustainable development requires climate change mitigation and thereby a fast energy transition to renewables. However, renewable power outputs are subjected to the weather variability and seasonal weather changes as those analyzed by the authors. They found that cloudiness during the rain season can reduce surface irradiance by more than 4% in the Yangtze-Huaihe River basin of China. This seems to be a small reduction but it translates into an equivalent drop in photovoltaic (PV) power generation, which leads in turn to a significant increase in the GHG emissions (that nowadays can amount up to about a quarter of million of tons, according to the authors). Their analysis is simply but powerful and necessary.

Answer:

We appreciate the Reviewer's invaluable suggestions and comments.

Your positive affirmation of our work prompted us to further improve the quality of this work. The manuscript has been carefully revised according to the Reviewer's comments. The detailed replies are presented one by one as follows.

However, I have major reservations regarding the first important result: the drop in the surface irradiance due to the predominantly overcast conditions during the rain season. The authors argue that this reduction is about 4%, which seems to be very low compared to the reported drop in surface irradiance associated to the Indian monsoon, for example, which is on average about 30%. Heavy clouds conditions can attenuate the solar radiation up to 80% such that an average attenuation of 4% during the rain season seems to be too low. It is also low compared with the effect on the solar potential of aerosols in China (see Sweerts, B., Pfenninger, S., Yang, S., Folini, D., Van der Zwaan, B. and Wild, M., 2019. Estimation of losses in solar energy production from air pollution in China since 1960 using surface radiation data. *Nature Energy*, 4(8), pp.657-663).

Answer:

We appreciate the Reviewer's invaluable suggestions and comments.

First of all, the 4% reduction in the initial manuscript is the average value of the entire affected region during the plum rain periods, and the peak reduction in some areas such as Shanghai can reach more than 10%. **In the revised manuscript, we have adopted the new indicator suggested by the reviewer for measuring plum rain's effects on all-sky surface irradiance (SI) and corresponding PV generation.** The new indicator is defined in the next response.

With the new indicator, we have re-analyzed and found that plum rain could obviously weaken SI in the affected region during its duration, **with a peak drop of more than 20% at the most affected locations.** Detailed information for plum rain effects on the SI can be seen in Figs. 1,2.

Also, we found that the incremental CO₂ emissions (ICEs) caused by plum rain were underestimated in the initial manuscript. Using the new indicator suggested by the reviewer, we found that the ICEs under different years in the revised manuscript are much higher than those in the initial manuscript. **For example, the peak ICEs in 2040 are 1.44 megatons in the initial manuscript, while these under the same conditions in the revised manuscript are 5.16 megatons,** with an increase of 3.6 times. Detailed information can be seen in Fig. 3.

Fig. 1 Plum rain effects on the SI in R1. a The 52-week SI within a year, where R1 denotes the affected region and R2 denotes the unaffected surrounding region. **b** Impact degree of plum rain on the SI from the 25th week to the 29th week. θ_p^{MEAN} in **b** denotes the impact degree of plum rain on the SI in each province or municipality.

Fig. 2 Spatial distributions of the impact degree within R1. The region enclosed within the red dashed line is R1. The region between the red and blue dashed lines is R2. θ^{MEAN} denotes the impact degree of plum rain on the SI at a given location.

Fig. 3 Plum rain effects on the ICEs from 2020 to 2050. The error bar defines the range of the ICEs based on the generation and demand change trends fluctuating by 25% from 2030 to 2050.

The authors have computed the drop in the surface irradiance by using the ratio between the all-sky surface irradiance and the top-of-atmosphere solar irradiance. However, the latter does not include the attenuations due to light-absorbing aerosols (that in the case of industrial China can hardly be ignored). The effect of the cloudiness on the surface irradiance is often computed by using the so-called cloud modification factor (CMF) taken as equal to the ratio between the all-sky surface irradiance and the clear-sky surface irradiance. The latter does include the effect of aerosols and it is of course a little bit lower than the top-of-atmosphere solar irradiance. The clear-sky surface irradiance is not always available in reanalysis datasets as those used by the authors. In atmospheric sciences, a frequent approach is to estimate the clear-sky surface irradiance by using a radiative transfer model. However, in this case, I suggest to the authors an empirical approach: to validate their estimation of the cloud-related surface irradiance drop but simply comparing the surface irradiance (in w/m²) averaged over the region most affected by the rain season and the surface irradiance (in w/m²) averaged over neighbor region less affected by the rain season. My understating is that they indirectly did this anyway creating the regions that they call R1 and R2. However, rather the unnecessarily complicated formulation presented by the authors, I would like to see a plot, simply showing changes through the year in the monthly all-sky surface irradiance average over a reasonable long period (1991-2020 for example), for region R2 and regions R1. Both curves should separate during the rain season exposing the effect of the more abundant clouds.

Answer:

We appreciate the Reviewer's invaluable suggestions and comments.

We have made significant modifications to assess the plum rain's impacts, as follows.

First, the unaffected surrounding region (R2) has been revised. According to the national standard (GB/T 33671-2017) of "plum rain monitoring indices" implemented on 2017-12-01, the plum rain-affected area covers the Yangtze-Huaihe River basin, including Jianghuai District, the Yangtze River Middle and Lower Reaches and Jiangnan District, with an affecting area of nearly 500000 square kilometres.^[R1] On this basis, we further extend the affected area by 200 km, obtaining an unaffected surrounding region (R2) with a similar size. The average value of SI in R2 is taken as the criterion not affected by plum rain. The boundaries of R1 and R2 can be seen in Supplementary Fig. 1.

Supplementary Figure 1 Boundaries of the affected region (R1) and the unaffected surrounding region (R2). The region enclosed by the red dashed line is R1. The region between the red dashed line and the blue dashed line is R2.

Second, instead of using SI of selected cities, we collect the SI in R1 and R2 with a spatial resolution of $1/2^\circ$ latitude by $2/3^\circ$ longitude from 1980 to 2020 derived from NASA's MERRA-2.^[R2] This method can avoid the one-sided impact of the uneven selection of representative cities on the evaluation of plum rain's effects. Detailed modifications can be seen in Fig. 1c.

Third, we assess the impact degree of plum rain on SI through an empirical approach shown in formula (1) as suggested by the reviewer. The impact degree of plum rain on SI at a given location in R1 can be calculated by the ratio of the difference between the $I_{SI}^{MEAN}(c, w)$ and $I_{SI}(c, w)$ to the $I_{SI}^{MEAN}(c, w)$, where $I_{SI}^{MEAN}(c, w)$ is the mean SI at w -th week in R2, and $I_{SI}(c, w)$ is the SI at a given location c in R1. M_{R1} is the set of spatial resolutions in R1, and Φ^W is the set of weeks during the plum rain period.

$$\theta^{MEAN}(c, w) = \frac{I_{SI}^{MEAN}(w) - I_{SI}(c, w)}{I_{SI}^{MEAN}(w)}, \forall c \in M_{R1}, w \in \Phi^W \quad (1)$$

Fourth, the reviewer suggested that we show changes through the year in the monthly all-sky mean surface irradiance over a reasonably long period. Considering that the monthly data cannot accurately express the impact of plum rain on a smaller time scale, we have averaged the 41-year data from 1980 to 2020 to one year and then showed the SI for this year in R1 and R2. The result is shown in Fig. 1a. **As indicated, the SI in R1 shows a significant downward trend compared with that in R2 during the plum rain period. The largest decline occurred in the 26th week, with a drop of 27.6 W/m².** Outside the plum rain period, the SI curves of the two regions overlapped very high. From the above observations, it is clear that plum rain can obviously reduce SI in R1.

There is another major point. What about the projections regarding the aerosol load? The light-absorbing aerosol load currently affecting China can hardly be ignored. By mid-century however, it is expected that this heavy aerosol load will significantly fall. Reverting back to 1960s radiation levels in China could yield a 12–13% increase in electricity generation (Sweerts, B., Pfenninger, S., Yang, S., Folini, D., Van der Zwaan, B. and Wild, M., 2019. Estimation of losses in solar energy production from air pollution in China since 1960 using surface radiation data. *Nature Energy*, 4(8), pp.657-663). For projecting changes in the solar potential in China, you need to consider the expected changes in the aerosol load.

Answer:

We appreciate the Reviewer's invaluable suggestions and comments.

As noted by the reviewer, anthropogenic aerosol emissions and changes in cloud cover affect solar radiation in China, which makes the PV potential decreased on average by 11–15% between 1960 and 2015.^[R3] Consider the reduction of fossil energy and the popularization of renewable energy in the future, this phenomenon is expected to be improved. Therefore, the above 10% impact degree of anthropogenic aerosol emissions needs to be considered in our model.

Following the study by Sweerts et al in Reference [R3], we assume that China's anthropogenic aerosol emissions in 2060 are consistent with 1960, that is, by improving solar radiation to achieve the 11–15% increase in PV potential. Although China's CO₂ emissions will still increase and the peak is expected to be in 2030, the implementation of ultra-low emissions standards for coal-fired power plants^[R4] and electric power alternation such as Electric Heating Policies (EHPs)^[R5] have improved the clean use of coal. Therefore, we believe that from 2020 to 2060 is a process of gradual reduction of anthropogenic aerosol emissions. **We suppose that the incremental PV generations in 2030, 2040, 2050, and 2060 are 1.03, 1.06, 1.09, and 1.12 of these in 2020, respectively.**

The formulas of PV power generation have been revised as

$$0 \leq P_{l_1}^{\text{PV}}(t) \leq (1 + \gamma^{\text{aerosol}}) \rho_{\text{R1}}^{\text{PV}}(t) C_{l_1}^{\text{PV}}, \forall l_1 \in \Phi_{\text{R1}}^{\text{PV}}, t \in \Phi^{\text{T}} \quad (2)$$

$$0 \leq P_{l_2}^{\text{PV}}(t) \leq (1 + \gamma^{\text{aerosol}}) \rho_{\text{R2}}^{\text{PV}}(t) C_{l_2}^{\text{PV}}, \forall l_2 \in \Phi_{\text{R2}}^{\text{PV}}, t \in \Phi^{\text{T}} \quad (3)$$

Constraints (11)-(12) define the lower and upper limits of PV power generation in R1 and R2, respectively, where γ^{aerosol} is the percentage increase of PV potential caused by anthropogenic aerosol emissions.

Looking forward for a revised version.

I also have couple of minor suggestions

- 1) The numbers in the scales/axes in most of figures (Fig. 2 for example) are too small and can hardly be read.
- 2) The terms adopted in the paper for referring to the all-sky surface irradiance, the clear-sky surface irradiance, the cloud modification factor (CMF) are not conventional. Please change them.

Answer:

We appreciate the Reviewer's invaluable suggestions and comments.

- 1) The numbers in the scales/axes in all figures have been enlarged to make them easily read.
- 2) The description of solar irradiance in the initial manuscript has been changed to all-sky surface irradiance, the clear-sky surface irradiance, the cloud modification factor (CMF).

All the simulation results have been accordingly revised.

In the end, the authors would like to thank the Reviewer for all the invaluable suggestions and comments on this paper. It is your kind help that makes our work better.

References

- R1 National Climate Center. Meiyu monitoring indices. <http://www.gb688.cn/bzgk/gb/newGbInfo?hcno=9E7EA80586ECADCBF5A65220782EDF1D> (2017)
- R2 Gelaroa, R., McCarty, W., and Suárez, M. J. et al. The modern-era retrospective analysis for research and applications, version 2 (MERRA-2). *J. Clim.* **30**, 5419–5454 (2017).
- R3 Sweerts, B., Pfenninger, S., Yang, S. et al. Estimation of losses in solar energy production from air pollution in China since 1960 using surface radiation data. *Nat. Energy* **4**, 657–663 (2019).
- R4 Ministry of Ecology and Environment of the People's Republic of China, National Development and Reform Commission & National Energy Administration Work. *Plan of Full Implementing Ultra-Low Emission Policy and Energy Saving Transformation for Coal-Fired Power Plants* (Ministry of Ecology and Environment of the People's Republic of China, 2015).
- R5 Wang, J., Zhong, H., Yang, Z. et al. Exploring the trade-offs between electric heating policy and carbon mitigation in China. *Nat. Commun.* **11**, 6054 (2020).

Reviewer #3 (Remarks to the Author):

This paper discusses the role of "plum rain" on solar power potential in China and potential solutions to balance electricity generation during these times. The idea of this paper, on its surface, is interesting and highly topical as countries gear towards more decarbonization commitments. The authors clearly committed significant work to this project, but there are a number of details missing that I think, at present, make the paper unfit for acceptance. With significant improvements to the manuscript, I think that this study could become more appropriate for Nature Communications.

Answer:

We appreciate the Reviewer's invaluable suggestions and comments.

Your positive affirmation of our work prompted us to further improve the quality of this work. The manuscript has been carefully revised according to the Reviewer's comments. **In particular, the Reviewer pointed out a number of details missing in the optimization model, which has been supplemented in the revised manuscript.** The detailed replies are presented one by one as follows.

Major comments:

1) Solar PV: I'm a bit confused as to how solar power potential was calculated in this paper. The authors reference Chen et al. (2019), but it appears that the temperature dependence of solar power electricity generation is not included in the calculations done here (which should be an important consideration in a paper that discusses the effects of climate change on solar power potential). Furthermore, more information is needed as to what solar power potential data are used for the Unit Commitment model. Are future projections of solar potential based on the average meteorological conditions from 2000-2019? I think that's reasonable, but should be justified in the text. Also, some of the analysis studies the data at the city-level, but isn't the reanalysis product too spatially coarse to analyze at such a granular level?

Answer:

We appreciate the Reviewer's invaluable suggestions and comments.

We apologize for not explaining the calculation method of solar power potential in the initial manuscript, which has been clarified in the revised manuscript. In view of the inability to accurately represent plum rain's impact on surface irradiance (SI) and solar power potential at the city level, we use sufficiently comprehensive data with a spatial resolution of $1/2^\circ$ latitude by $2/3^\circ$ longitude from 1980 to 2020, to observe the plum rain effects on the SI in the affected region. Detailed modifications are as follows:

The open-source Global Solar Energy Estimator (GSEE) model^[R1] on the www.renewables.ninja web platform is adopted to estimate PV capacity factors. According to the study by Pfenninger et al, **the power output of PV modules depends on the in-plane irradiance G and module temperature T_{mod} :**

$$P^{\text{PV}}(G, T_{\text{mod}}) = P_{\text{STC}}^{\text{PV}} \cdot \frac{G}{G_{\text{STC}}} \cdot \eta_{\text{rel}}(G', T') \quad (1a)$$

where $P_{\text{STC}}^{\text{PV}}$ is the power output at standard test conditions (STC) with in-plane irradiance G_{STC} of $1000\text{W}/\text{m}^2$ and module temperature $T_{\text{mod_STC}}$ of 25°C . The hourly instantaneous relative efficiency η_{rel} , depending on the instantaneous irradiance and temperature, is given by^[R2]

$$\eta_{rel}(G', T') = 1 + k_1 \ln G' + k_2 (\ln G')^2 + T' \left[k_3 + k_4 \ln G' + k_5 (\ln G')^2 \right] + k_6 T'^2 \quad (1b)$$

where $G' = \frac{G}{G_{STC}}$ and $T' = T_{mod} - T_{mod_STC}$ are normalized parameters to STC values. k_1 - k_6 are coefficients determined by experimental data. T_{mod} can be further calculated by the ambient temperature and irradiation G as follows:

$$T_{mod} = T_{amb} + c_T G \quad (1c)$$

where c_T represents how much PV module is heated by irradiation G .

The in-plane irradiance G includes the direct and diffuse plane irradiance (G_{dir} and G_{dif}), which can be computed from the global direct and diffuse irradiance (I_{dir} and I_{dif}) by

$$G_{dir} = \frac{I_{dir} \cdot \cos(\alpha)}{\cos\left(\frac{\pi}{2} - a_s\right)} \quad (1d)$$

$$G_{dif} = I_{dif} \cdot \frac{1 + \cos(\Sigma)}{2} + a \cdot (I_{dir} + I_{dif}) \cdot \frac{1 - \cos(\Sigma)}{2} \quad (1f)$$

where α is the plane incidence angle, calculated by (1g) with a fixed tilt angle. a_s is the sun azimuth angle, a is the surface albedo ($a=0.3$).

$$\alpha = \arccos\left(\sin(h) \cdot \cos(\Sigma) + \cos(h) \cdot \sin(\Sigma) + \cos(a_p - a_s)\right) \quad (1g)$$

where h is sun altitude, a_p is panel azimuth, and a_s is sun azimuth angle.

Σ is the plane tilt in degrees, calculated by

$$\Sigma = 1.3793 + lat \cdot (1.2011 + lat \cdot (-0.014404 + lat \cdot 0.000080509)) \quad (1h)$$

where lat is the latitude in degrees. Considering that the plum rain-affected areas occur near latitude 30°N, we set the optimum tilt angle of the fixed-tilt system to 26.6 degrees, following the study by Chen et al.^[R3] In the end, the system loss is set to the default value of 0.1.

Using the above model, we take the points of each province with the spatial resolution of 1/2° latitude by 2/3° longitude shown in Supplementary Fig. 5, and calculate the hourly mean unit PV outputs $\rho_{R1}^{PV}(t)$ and $\rho_{R2}^{PV}(t)$ in the affected region and not affected region, respectively.

The calculated the hourly mean unit PV outputs are used for future projections of solar potential.

Supplementary Figure 5 Spatial resolutions of $1/2^\circ$ latitude by $2/3^\circ$ longitude to calculate PV capacity factor.

2) Unit commitment model assumptions: Implications of potential plum rain effects on the electricity grid are certainly interesting, but I think there is a lot of missing information for how the model is run. For instance, assumptions of cost for various technologies are one of the most important components of a unit commitment model. As far as I can tell, there is no information on costs in the text (except for a brief mention of three cost scenarios), nor is there any justification for the costs assumed. This should, at the very least, be a significant component of the methods section. Technologies like wind and nuclear are included in the methods section, but are not mentioned in the results – are these included in the unit commitment modeling?

Answer:

We appreciate the Reviewer’s invaluable suggestions and comments.

As the Reviewer pointed out, some key information for how the model is run is not given in the initial manuscript, and these have been completely given in the Methods or Supplementary Information in the revised version. It should be noted that all information on costs comes from recently published authoritative references or public reports released by authoritative institutions.

First, the Li-ion storage with durations up to 4 h at rated power^[R4] is chosen to help PV integration in power systems, and the energy capacity cost is set to 1380 ¥/kWh^[R5]. The penalty factor regarding solar power and load curtailments is set to a sufficiently large value of 100 ¥/kg to avoid solar power and load curtailments as much as possible. The fuel costs of coal-fired and gas-fired generators are 0.37 and 0.65 ¥/kWh^[R6] respectively, and the corresponding mission factors are 1.22945 and 0.3756 kg/kWh^[R7]. Based on similar nameplate capacity, six groups are clustered for coal-fired generators and four groups are clustered for gas-fired generators (see Supplementary Table 3), with the numbers of all groups in each province collected from the China Electric Power statistical yearbook 2020^[R8]. The startup and shutdown

cost of a generating unit are assumed to be proportional to the capacity.^[R9] For example, a 500-MW unit has a startup and shutdown cost of 500000 ¥. In addition, we evaluate the carbon reduction effects of the demand response program, CCUS, and long-duration storage based on current and future potential techno-economic parameters.

Since we focus on the impact of plum rain on PV generation, followed by the study of Wang et al^[R10], we treat wind and nuclear in the unit commitment model as known values, not optimization variables in the unit commitment model. Therefore, they are not mentioned in the results. Also, we have clarified all the optimization variables in the decision variable set **X** of the objective function (2) in the revised manuscript.

Supplementary Table 3. Economic and technical parameters of clustered power plant types.^[R8]

Plant type	Capacity (MW)	Startup/shutdown time (h)
Coal generator	≤ 1000	16
	$600 \leq \text{and} < 1000$	8
	$300 \leq \text{and} < 600$	7
	$200 \leq \text{and} < 300$	6
	$100 \leq \text{and} < 200$	5
	$6 \leq \text{and} < 100$	2
Natural gas generator	$300 \leq \text{and} < 600$	7
	$200 \leq \text{and} < 300$	6
	$100 \leq \text{and} < 200$	5
	$6 \leq \text{and} < 100$	2

* The startup and shutdown cost of generating unit is assumed to be proportional to the capacity^[R9]. For example, a 500-MW unit has a startup and shutdown cost of 500000 ¥.

3) Mitigation methods: Similarly to my prior comments, I think there needs to be many more details for this section. What's the difference between Hydrogen1 and Hydrogen2 (if any at all besides discharge time)? What's the source of the hydrogen; is this grey, blue or green hydrogen? What are potential methods of demand response changes? It would also help to clarify the captions for figures 3 and 4 which I found to be confusing.

Answer:

We appreciate the Reviewer's invaluable suggestions and comments.

We have modified the representative mitigation methods, and added the following selected principles in the revised manuscript.

To offset the incremental CO₂ emissions caused by plum rain, we can adopt measures not to increase the output of thermal generators, or reduce the CO₂ emissions of thermal generators when filling the missing amount of PV generation caused by plum rain. First, we consider the power conversion of CG-to-NG (C2N), that is, increasing the output of NGs and reducing the output of CGs to make the CO₂ emissions of power systems reach a level that does not affected by plum rain. On the basis of C2N, three promising technologies, including carbon capture, utilization and storage (CCUS), demand response (DR), and long-duration (LD) storage, are considered. Since CG has a higher carbon emission factor than NG, CCUS is deployed to some CGs to reduce the CO₂ emissions of thermal generators from source. The incentive-based DR

program, which can offer payments for users to reduce their electricity usage during periods of system need or stress in a long-term,^[R11] is adopted in the optimization model. LD storage can help shift energy during multi-day periods of supply and demand imbalance, and thus can be used to store/release electric energy before/during the plum rain period. Here, hydrogen storage is selected as a very promising technology where energy storage capacity can be designed fully independent of power capacity.^{[R12], [R13]} **Since we did not consider hydrogen storage with a fixed power-to-capacity ratio, we have not distinguished Hydrogen1 and Hydrogen 2 in the revised manuscript.**

We assume that green hydrogen is produced by surplus electric power from undispachable renewables, and is stored by LD storage before the rainy season. We have clarified the green hydrogen in the revised manuscript.

As potential methods for reducing power load, DR programs can be divided into two major programs: time-based DR programs, and incentive-based DR programs.^[R11] Both types of DRs are currently under operation in many ISOs around the world. The time-based DR programs are established to overcome flat or averaged electricity pricing flaws. Time-of-use tariffs, critical-peak pricing, and real-time pricing are the three well-known time-based DR programs. The incentive-based DR programs offer payments for customers to reduce their electricity usage during periods of system need or stress. Different types of incentive-based programs span over long-term to mid-term, short-term, and even real-time offered programs. **Therefore, the incentive-based DR program is adopted in our optimization model.** Also, we have given the levelized cost of CO₂ mitigation and corresponding compensation energy under different DR compensation costs and powers in Fig. 5 and Supplementary Fig. 4 to study the impact of potential changes on the carbon reduction effect of the DR program.

Fig. 5 LCCM and compensation energy under the C2N+DR, C2N+CCUS, and C2N+LD pathways considering the different techno-economic parameters. a LCCM of C2N+DR under different DR compensation cost and DR power levels. **b** LCCM of C2N+CCUS under different CCUS costs and efficiencies. **c** LCCM of C2N+LD under different power and energy capacity cost levels. **d** Compensation energy for C2N+DR under different DR compensation cost and DR power levels. **e** Clean energy for C2N+CCUS under different CCUS costs and

efficiencies. f Net released energy for C2N+LD under different power and energy capacity cost levels.

Minor comments:

L12: Should explain what "plum rain" is in the abstract.

Answer:

We have explained what "plum rain" is in the abstract, as follows:

As a typical climate occurred in the Yangtze-Huaihe River basin of China, plum rain may reduce PV generations by covering surface irradiance (SI) in the affected region.

L51: I think this is meant to be "2060" instead of "3060".

Answer:

"3060" is China's latest carbon reduction target, that is, to achieve a carbon dioxide emission peak by 2030 and achieve carbon neutrality by 2060. We have changed it into the "3060 target" in the revised manuscript.

L82-83: I think this discussion of climate change is confusing because it's not actually being investigated in the paper.

Answer:

The discussion of "climate change" has changed to "typical climate".

L108-109: This should be elaborated further here as to why that's the case (it makes sense after reading the rest of the paper, but it was unclear at the time of reading).

Answer:

We have hereby declared that plum rain can obviously reduce PV generations via lowering surface irradiance in the affected areas, and thus increase CO₂ emissions of power systems.

L130: The trend doesn't seem statistically significant

Answer:

We have deleted these contents in the revised manuscript, considering that the trend doesn't seem statistically significant.

L130-132: How can you conclude that 2020 has a $\Delta\theta$ above 7%? Is this not shown in the figure?

Answer:

Fig. 1b showing the effect in the different years has been deleted. Also, the meteorological data from 2000 to 2019 has been expanded from 1980 to 2020.

L144-145: The caption should include the definition of $\Delta\theta_{\text{TIME}}$.

Answer:

We have added the definition that appeared in the caption of the figure.

θ_p^{MEAN} in Fig. 1b and θ^{MEAN} in Fig.2 denote the impact degree of plum rain on SI at each province/municipality and a given location, respectively.

Figure 1, panel c: It would be helpful to identify R1 and R2 in the figure so the reader doesn't

have to look through the caption to find what each is referring to.

Answer:

As noted by the reviewer, we have directly identified R1 and R2 in the figure.

Supplementary Figure 1 Boundaries of the affected region (R1) and the unaffected surrounding region (R2). The region enclosed by the red dashed line is R1. The region between the red dashed line and the blue dashed line is R2

L168: What are the numbers for PV capacity quoted here? Do these represent the projected 2050 PV capacity or something else?

Answer:

The numbers for PV capacity quoted here is the projected PV capacity in the affected region from 2020 to 2050, and is predicted based on the China Energy & Electricity Outlook published by State Grid Energy Research Institute.

L190-191: How can this be concluded? The climate data used in the paper goes from 2000-2019, so how can conclusions be made about the PV output from 2020-2050? Is there information missing?

Answer:

As noted by the reviewer, we have declared that the calculated hourly mean unit PV outputs from 1980 to 2020 are used for future projections of solar potential.

L214-215: Should include discussion of potential methods for reducing power load.

Answer:

We have discussed the potential methods for reducing power load. Detailed modification can be seen in the above response and revised manuscript.

L216: Capacity is units for GW, not GWh. Also, what does it mean for DR energy capacity to

reach 2900 GWh? Is that saying how much electricity you would need to save by reducing power load?

Answer:

We have clarified the power and energy of the DR program in the revised manuscript. DR power is measured as a percentage of maximum load. DR energy is the compensating energy of the DR during the entire rainy season.

Table 1: Missing "1" and "2" for hydrogen1 and hydrogen2 in the table. Though I'm still not sure what the difference is between the two and why it's labelled as such.

Answer:

Thanks to the reviewers for pointing out this error. The hydrogen storage is not divided into hydrogen1 and hydrogen2 in the revised manuscript.

Figure 3: I found this figure highly confusing. Does each dot represent a scenario run in the unit commitment model? What do the units of GWh mean for DR and CCUS?

Answer:

Fig. 3 has been revised as the new Fig. 4 in the revised manuscript. Each techno-economic parameter represents a scenario run in the unit commitment model. The units of GWh for DR and CCUS means the compensating energy by DR and clean energy produced by CG with CCUS during the plum rain period.

L275-276: There needs to be discussion of how the prices assumed for the paper.

Answer:

The three cost scenarios in the initial manuscript have been deleted. In the revised manuscript, we have studied the impact of various technologies on carbon reduction under a wider range of techno-economic parameters based on the current and future technological levels.

Figure 4: There seems to be a lot of missing information for this figure. What does each of these subplots show and why is the y axis different for each technology studied? Also, what does the time on the x axis mean in the context of this figure?

Answer:

Fig. 4 in the initial manuscript has been deleted. As an alternative, the new Fig. 5 shows the power balance under different pathways, where the x-axis is the 168 hours during a week period from July 2 to July 8, and the y-axis shows the hourly power of different generators.

L345: this should say "...CO2 emissions from the electricity sector...", specifically.

Answer:

We have added "from the electricity sector" in the above sentence.

L349-350: I think this point about renewables taking up land is a bit of an overstatement, especially for China where the optimal wind resources are in Inner Mongolia and offshore.

Answer:

We have deleted the above inappropriate description.

All the simulation results have been accordingly revised.

In the end, the authors would like to thank the Reviewer for all the invaluable suggestions and comments on this paper. It is your kind help that makes our work better.

References

- R1 Pfenninger, S. & Staffell, I. Long-term patterns of European PV output using 30 years of validated hourly reanalysis and satellite data. *Energy* **114**, 1251–1265 (2016).
- R2 Huld, T., Gottschalg, R., Beyer H. G., Topič, M. Mapping the performance of PV modules, effects of module type and data averaging. *Sol. Energy* **84(2)**, 324-338 (2010).
- R3 Chen, S., Lu, X., Miao Y., et al. The potential of photovoltaics to power the belt and road initiative. *Joule* **3**, 1895–1912 (2019).
- R4 Albertus, P., Manser, J. S., and Litzelman, S. Long-duration electricity storage applications, economics, and technologies. *Joule* **4**, 21–32, (2020).
- R5 He, G., Michalek J., Kar, S., Chen, Q., Zhang, Da., and Whitacre, J. F. Utility-scale portable energy storage systems. *Joule* **5**, 1–4 (2021).
- R6 Chen, X., McElroy M. B., and Kang, C. Integrated Energy Systems for Higher Wind Penetration in China: Formulation, Implementation, and Impacts. *IEEE Trans. on Power Syst.* **33**, 1309-1319 (2018).
- R7 Chen, X., Zhang, H., Xu, Z. et al. Impacts of fleet types and charging modes for electric vehicles on emissions under different penetrations of wind power. *Nat. Energy* **3**, 413–421 (2018).
- R8 China Electricity Council. China Electric Power statistical yearbook 2020. http://www.stats.gov.cn/tjsj/tjcbw/202103/t20210329_1815748.html
- R9 Zhong, H., Xia, Q., Chen, Y., Kang, C. Energy-saving generation dispatch toward a sustainable electric power industry in China. *Energy policy* **83**, 14-25 (2015).
- R10 Wang, J., Zhong, H., Yang, Z. et al. Exploring the trade-offs between electric heating policy and carbon mitigation in China. *Nat. Commun.* **11**, 6054 (2020).
- R11. Parvania, M. and Fotuhi-Firuzabad, M. Demand response scheduling by stochastic SCUC. *IEEE Trans. Smart Grid* **1**, 89–98 (2010).
- R12 Sepulveda, N.A., Jenkins, J.D., Edington, A. et al. The design space for long-duration energy storage in decarbonized power systems. *Nat. Energy* **6**, 506–516 (2021).
- R13 Albertus, P., Manser, J. S., and Litzelman, S. Long-duration electricity storage applications, economics, and technologies. *Joule* **4**, 21–32, (2020).

Reviewer comments: - -

Reviewer #2 (Remarks to the Author):

I am happy to see that the authors adopted my suggestion regarding their first important result: the drop in the surface irradiance due to the predominantly overcast conditions during the rain season. The authors now estimate the cloud-related surface irradiance drop by comparing the surface irradiance (in w/m^2) averaged over the region most affected by the rain season and the surface irradiance (in w/m^2) averaged over neighbor region less affected by the rain season. The estimation appears to be now more realistic.

The authors have also addressed another major point by considering the effect of the expected aerosol changes in China. My minor suggestions regarding the figures were also addressed.

I have no further suggestions.

Reviewer #3 (Remarks to the Author):

I think the authors have done a great job addressing my concerns with the paper and believe it is ready for acceptance.

Reply to the comments

Manuscript ID: NCOMMS-21-16660

Title: Assessment of plum rain's impact on power system emissions in Yangtze-Huaihe River basin of China

Reviewer #2 (Remarks to the Author):

I am happy to see that the authors adopted my suggestion regarding their first important result: the drop in the surface irradiance due to the predominantly overcast conditions during the rain season. The authors now estimate the cloud-related surface irradiance drop by comparing the surface irradiance (in w/m^2) averaged over the region most affected by the rain season and the surface irradiance (in w/m^2) averaged over neighbor region less affected by the rain season. The estimation appears to be now more realistic.

The authors have also addressed another major point by considering the effect of the expected aerosol changes in China. My minor suggestions regarding the figures were also addressed.

I have no further suggestions.

Answer:

We appreciate the Reviewer's invaluable suggestions and comments. Your positive affirmation of our work prompted us to further improve the quality of this work.

Thank you again for your invaluable contribution to the publication of this work in Nature Communications.

Reviewer #3 (Remarks to the Author):

I think the authors have done a great job addressing my concerns with the paper and believe it is ready for acceptance.

Answer:

We appreciate the Reviewer's invaluable suggestions and comments. Your positive affirmation of our work prompted us to further improve the quality of this work.

Thank you again for your invaluable contribution to the publication of this work in Nature Communications.